# PF-GNN: Differentiable particle filtering based approximation of universal graph representations

**Mohammed Haroon Dupty**[1]**, Yanfei Dong**[1,2]**& Wee Sun Lee**[1]
[1]School of Computing, National University of Singapore      [2]PayPal Innovation Lab
{dmharoon,dyanfei,leews}@comp.nus.edu.sg

## ABSTRACT

Message passing Graph Neural Networks (GNNs) are known to be limited in expressive power by the 1-WL color-refinement test for graph isomorphism. Other more expressive models either are computationally expensive or need preprocessing to extract structural features from the graph. In this work, we propose to make GNNs universal by guiding the learning process with exact isomorphism solver techniques which operate on the paradigm of *Individualization and Refinement* (IR), a method to artificially introduce asymmetry and further refine the coloring when 1-WL stops. Isomorphism solvers generate a search tree of colorings whose leaves uniquely identify the graph. However, the tree grows exponentially large and needs hand-crafted pruning techniques which are not desirable from a learning perspective. We take a probabilistic view and approximate the search tree of colorings (*i.e.* embeddings) by sampling multiple paths from root to leaves of the search tree. To learn more discriminative representations, we guide the sampling process with *particle filter* updates, a principled approach for sequential state estimation. Our algorithm is end-to-end differentiable, can be applied with any GNN as backbone and learns richer graph representations with only linear increase in runtime. Experimental evaluation shows that our approach consistently outperforms leading GNN models on both synthetic benchmarks for isomorphism detection as well as real-world datasets.

## 1 INTRODUCTION

In recent years, Graph Neural Networks (GNNs) have emerged as learning models of choice on graph structured data. GNNs operate on a message passing paradigm (Kipf & Welling, 2016; Defferrard et al., 2016; Veličković et al., 2017; Gilmer et al., 2017), where nodes maintain latent embeddings which are updated iteratively based on their neighborhood. This way of representation learning on graphs provides the necessary inductive bias to encode the structural information of graphs into the node embeddings. The process of message passing in GNNs is equivalent to vertex color-refinement procedure or the 1-dimensional Weisfeiler-Lehman (WL) test used to distinguish non-isomorphic graphs (Xu et al., 2018; Morris et al., 2019). Consequently, GNNs suffer from the limitations of 1-WL color-refinement in their expressive power.

In each step of 1-WL color-refinement, two vertices get different colors if the colors of neighboring vertices are different. The procedure stabilizes after a few steps when the colors cannot be further refined. Due to the symmetry in graph structures, certain non-isomorphic graphs induce same colors upon 1-WL refinement. Higher-order WL refinement and their neural $k$-GNN versions break some of the symmetry by operating on $k$-tuples of nodes. They are more expressive but require exponentially increasing computation time and hence, are not practical for large $k$. Motivated by the fact that a fully expressive graph representation learning model should be able to produce embeddings that can distinguish any two non-isomorphic graphs (Chen et al., 2019b), we turn to exact graph isomorphism solvers for better inductive biases in our learning algorithm.

Most of the practical graph isomorphism solvers use 1-WL in combination with the traditional technique of *individualization and refinement* (IR) (McKay & Piperno, 2014; Junttila & Kaski, 2011)

for coloring the graph. *Individualization* is the process of artificially introducing asymmetry by recoloring a vertex and thereby, distinguishing it from the rest of the vertices. *Refinement* refers to 1-WL refinement which can propagate this information to recolor the rest of the graph. The two graphs shown in Fig. 1 are not distinguishable after 1-WL refinement but induce different colorings after one IR step. The IR process is repeated for each refinement until every vertex gets a unique color. However, in order to maintain permutation-invariance, whenever a vertex is individualized, other vertices that have the same color need to individualized as well and thereafter refined. This process generates a search tree with colorings as nodes, and can grow exponentially in worst case.

In this work, we propose to learn graph representations with the inductive bias of the search tree generated by the graph-isomorphism solvers. However, generating the complete search tree is computationally expensive. Isomorphism-solvers prune the search tree by detecting automorphisms on the fly as they generate the tree. Nevertheless, detecting automorphisms is non-trivial from the perspective of end-to-end discriminative learning and hence, we need to approximate it. To this end, we first define a universal neural graph representation based on the search tree of colorings. Then we take a probabilistic view and approximate it by sampling multiple paths from root to leaves of the search tree. We observe that the process of sampling a vertex, individualizing it and the subsequent refinement resembles the state transition in sequential state estimation problems.Hence, we model our sampling approach on the principled technique of Sequential Monte Carlo or Particle Filtering.

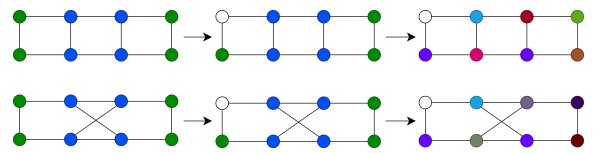

Figure 1: Two 1-WL equivalent graphs with different colorings after one step of individualization and refinement.

We introduce Particle Filter Graph Neural Networks (PF-GNN), an end-to-end learnable algorithm which maintains a weighted belief distribution over a set of $K$ graph colorings/embeddings. With each step of IR, PF-GNN transitions to a new set of embeddings. It then updates the belief by re-weighting each particle with a discriminatively learned function that measures the quality of the refinement induced after the transition. With this inductive bias, the belief evolves over time to be more discriminative of the input graph. After a few steps, we can use the belief along with the embeddings to generate the final representation of the graph. Our approach is simple, efficient, parallelizable, easy to implement and can learn representations to distinguish non-isomorphic graphs beyond 1-WL GNNs. We evaluate PF-GNN over diverse set of datasets on tasks which are provably not learnable with 1-WL equivalent GNNs. Furthermore, our experiments on real-world benchmark datasets show the strong performance of PF-GNN over other more expressive GNNs.

## 2 RELATED WORK

It was established that GNNs are limited in expressive power, and cannot go beyond 1-WL test for graph isomorphism by Xu et al. (2018) and Morris et al. (2019). Later analysis of GNN has shown other limits of GNNs like counting substructures and detecting graph properties (Arvind et al., 2020; Chen et al., 2020; Loukas, 2019; Dehmamy et al., 2019; Srinivasan & Ribeiro, 2019). Chen et al. (2019b) further formalizes the intuition that there is equivalence between learning universal graph representations and solving the graph isomorphism problem. Thereafter, many models have been proposed that are more expressive than 1-WL. Prominent ones are $k$-GNNs and their equivalent models (Morris et al., 2019; Maron et al., 2019; Vignac et al., 2020; Morris et al., 2020), but they are difficult to scale beyond 3-WL. Other methods need to preprocess the graph to find substructures (Bouritsas et al., 2020; Li et al., 2020), which are not task-specific and may incur more computation time.

Another way to improve expressivity of GNNs is to introduce randomness (You et al., 2019; Sato et al., 2020; Abboud et al., 2020; Zambon et al., 2020). However, adding uniform randomness interferes with the learning task, and hence these models have not shown good performance on real-world datasets. In PF-GNN, randomness is controlled as only a subset of nodes are sampled. Furthermore, since all components are learned discriminatively, the representation is tuned towards the end task. Our approach of learning with particle filter updates is inspired from recent works which make particle filters differentiable (Karkus et al., 2018; Ma et al., 2019; 2020).

## 3 PRELIMINARIES

Let $\mathbf{G}_n$ be the set of all graphs of $n$ vertices with vertex set $\mathcal{V} = \{1, 2, \ldots, n\}$ and edge set $\mathcal{E}$, Isomorphism between any two graphs $\mathcal{G}, \mathcal{H} \in \mathbf{G}_n$ is a bijection $f : \mathcal{V}_{\mathcal{G}} \to \mathcal{V}_{\mathcal{H}}$ such that $(u, v) \in \mathcal{E}_{\mathcal{G}} \iff (f(v), f(u)) \in \mathcal{E}_{\mathcal{H}}$. An automorphism of $\mathcal{G}$ is an isomorphism that maps $\mathcal{G}$ onto itself. One way of identifying non-isomorphic graphs is by generating unique *colorings* for graphs based on their structures in a permutation-invariant way and then, comparing them.

A colouring of the graph $\mathcal{G} \in \mathbf{G}_n$ is a surjective function $\pi : \mathcal{V} \to \mathbb{N}$, which assigns each vertex to a natural number. The number of colors is denoted by $|\pi|$. The set of vertices sharing the same color form a *color cell* in the coloring. We denote the set of colored vertices with $\boldsymbol{\pi} = \{p_1, p_2, \ldots, p_k\}$ where $p_i$ is a color cell. A coloring in which every vertex gets a distinct color is called a *discrete colouring i.e.* $|\pi| = n$. For any two colorings $\pi, \pi'$, we say that $\pi'$ is *finer than or equal to* $\pi$, written $\pi' \preceq \pi$, if $\pi(v) < \pi(w) \Rightarrow \pi'(v) < \pi'(w)$ for all $v, w \in V$. *i.e.* each cell of $\pi'$ is a subset of a cell of $\pi$, but the converse is not true. Coloring is also loosely called a *partition*, since it partitions $\mathcal{V}$ into cells. A coloring is an *equitable partition* when any two vertices of the same color are adjacent to the same number of vertices of each color.

1-dimensional Weisfeiler Lehman test is a simple and fast procedure to color graphs. It starts with the same initial color for all vertices. Then, it iteratively refines the coloring of the graph by mapping the the tuple of the color of a vertex and its neighbors to a distinct new color. *i.e* at step $t$, $\pi_{t+1}(v) = \text{HASH}\Big(\pi_t(v),\ \{\!\{\pi_t(u), u \in N(v)\}\!\}\Big)$, where $\{\!\{\}\!\}$ denotes a multiset and $N(v)$ is the set of adjacent vertices of $v$. The algorithm terminates when $\pi$ forms an equitable partition.

### 3.1 SEARCH TREE OF COLORINGS

Equitable coloring cannot be further refined due to the symmetry in the graph structure. To break the symmetry, exact isomorphism solvers employ the technique of *individualization-refinement* to generate further refined colorings. Individualization is the technique of breaking the symmetry in the graph by distinguishing a vertex with a new unique color. Once a vertex is individualized, 1-WL refinement is used to refine the coloring until a further refined equitable partition is reached. However, this comes at a cost. In order to maintain permutation-invariance, we have to individualize and refine all the vertices with the same color. As a result, we get as many refined colorings as the number of vertices with the chosen color. This procedure can be repeated iteratively for each of the refined coloring and the process takes shape of a rooted *search tree* of equitable colorings.

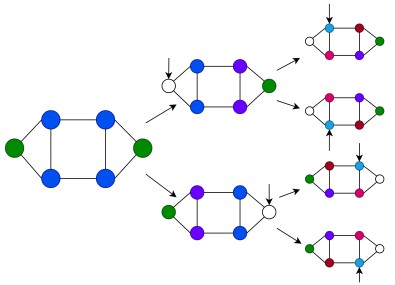

Figure 2: An example search tree of colorings generated by exact graph isomorphism solvers. Initial coloring is produced by 1-WL refinement. PF-GNN approximates the tree by sampling multiples paths from root to leaf.

The *search tree* has initial 1-WL coloring at the root. Then a non-singleton color cell called the *target cell* is chosen and all vertices of the target cell are individualized in parallel and thereafter, colorings are refined. The refined equitable colorings form the child nodes of the root. After sufficient repetitions, we get a set of discrete colorings at the leaves. This search tree is unique to the isomorphism class of graphs, *i.e.* all non-isomorphic graphs produce distinct search trees, and consequently, discrete colorings at the leaves. An example of search tree is shown in Fig. 2.

## 4 PROPOSED METHOD

In this section, we first define a universal representation of any $n$-vertex graph, then propose a practical approximation of the representation.

For any arbitrary $n$-vertex graph $\mathcal{G}$, we aim to learn its neural representation $f(\mathcal{G})$ that can uniquely identify $\mathcal{G}$. To achieve this, we can construct a computation graph which mimics the *search tree* of colorings. We propose to design a neural equivalent of the search tree. To do this, we can model the target-cell selector using a learnable function that outputs scalar scores on the vertices based

on their embeddings, and then individualize those which have the maximum score. Note that node embeddings in GNN are analogous to colors in 1-WL refinement. We can then adopt GNN to produce refined set of embeddings. If we have discrete embeddings after $T$ iterations, then $f(\mathcal{G})$ can be computed as,

$$f(\mathcal{G}) = \rho \left( \sum_{\forall \mathcal{I}} \psi(\mathcal{G}, \boldsymbol{\pi}_T^{\mathcal{I}}) \right) \tag{1}$$

where $\mathcal{I}$ is a sequence of vertices individualized iteratively ($\mathcal{I}$ identifies a single path from root to a leaf in the search tree) and the summation is over all such paths. $\boldsymbol{\pi}_T^{\mathcal{I}}$ is the discrete coloring (discrete embeddings) of $\mathcal{G}$ after individualizing and refining with vertices in $\mathcal{I}$. $\psi$ is an multiset function approximator and $\rho$ is an MLP. Theorem 1 presents the expressive power of $f(\mathcal{G})$.

**Theorem 1.** *Consider any $n$-vertex graph $\mathcal{G}$ with no distinct node attributes. Assume we use universal multiset function approximators for target-cell selection and graph pooling function $\psi$, and a GNN with 1-WL equivalent expressive power for color refinement, then the representation $f(\mathcal{G})$ in Eqn. 1 is permutation-invariant and can uniquely identify $\mathcal{G}$.*

Proof of Theorem 1 is provided in Appendix A.3.

### 4.1 PROBABILISTIC REPRESENTATION

The graph representation in Eqn. 1 is computationally expensive to obtain as the size of the search tree can grow exponentially with the number of vertices. Graph isomorphism solvers like Nauty and Traces (McKay & Piperno, 2014) prune the search tree by detecting automorphisms in order to run faster. To speed up, we instead propose to approximate the representations for the set of colorings at the leaves as suggested by Theorem 1 using sampling techniques, and approximate the pruning process by learning to down-weight certain subtrees.

By sequentially sampling the vertices to individualize and refine, we can get a sample set of paths/trajectories from root to leaf in the search tree. To facilitate the approximation, we use the expectation of the embeddings instead of the sum of the embeddings. Assume we need $T$ steps to generate discrete colorings. Let $\mathcal{I}$ be the sequence of vertices selected to individualize in one trajectory from root to leaf. Then, we have the following expectation

$$\tilde{f}(\mathcal{G}) = \frac{1}{|\Pi^{\mathcal{I}}|} \sum_{\mathcal{I} \in \Pi^{\mathcal{I}}} \psi(\mathcal{G}, \boldsymbol{\pi}_T^{\mathcal{I}}) = \mathbb{E}[\psi(\mathcal{G}, \boldsymbol{\pi}_T^{\mathcal{I}})] \tag{2}$$

where $\Pi^{\mathcal{I}}$ is the set of all paths in the search tree. In this case, we want the $\mathbb{E}[\psi(\mathcal{G}, \boldsymbol{\pi}_T^{\mathcal{I}})]$ to be different if the set of leaves is different; function $\psi$ with this property exists if the number of leaves and distinct leaf representations are bounded, e.g. if we limit the size of graphs we consider. To see this, observe that each distinct graph will have a distinct canonical embedding as one of the leaves of its search tree (McKay & Piperno, 2014), and $\psi$ can set one of the components of its output vector to be the indicator function of this canonical embedding. To approximate $\tilde{f}(G)$, we can initialize a set of $K$ colorings and in each step, sample a vertex to individualize and refine the colorings. After multiple such steps, we average the $K$ embeddings to obtain the final graph embedding. We bound the approximation error of the sampling process in the following theorem.

**Theorem 2.** *Assume we need $D$ dimensional embeddings for some $D > 0$, in order to uniquely represent any graph of $n$-vertices. Consider two $n$-vertex graphs $\mathcal{G}_1$ and $\mathcal{G}_2$ whose universal representations in Eqn. 2 are $f_1$ and $f_2$ after $T$ levels of the search tree. Let the max-norm distance between $f_1$ and $f_2$ be at least $d$ i.e. generating the full level-$T$ search tree for both graphs will separate them by $d$. Assume that the values in embeddings generated by the tree are strictly bounded in the range of $[-M, M]$. Then, with probability of at least $1 - \delta$, the approximate embeddings, generated by sampling $K$ paths from root to leaves, are separated in max-norm distance by $d \pm \epsilon$ provided the number of root to leaf paths sampled are at least*

$$K \in \mathcal{O}\left( \frac{M^2 \ln(\frac{4D}{\delta})}{\epsilon^2} \right)$$

The proof is provided in Appendix A.3. Further refinement of the method can be made in practice. First, we learn the function used to select the node for individualization so that it is adaptive to the

end-task. Additionally, the learned function may select different nodes with different probabilities, so the probability of each path down the tree is different. To speed up, practical solvers prune subtrees when they are unlikely to be helpful. We propose to approximate this process by learning how to de-emphasize certain subtrees using particle filter methods as described in the next section.

## 4.2 PARTICLE FILTER GRAPH NEURAL NETWORK

**Particle Filtering:** Particle filtering is a monte-carlo technique that sequentially estimates the posterior distribution of the state in non-linear dynamical systems when partial observations are made. Consider a process where we receive an observation $o_t$ at each time step when it transitions from state $x_t$ to a new state $x_{t+1}$ and we have a transition model $p(x_{t+1}|x_t)$, an observation model $p(o_t|x_t)$ and a sequence of observations $[o_1, o_2 \dots o_T]$. Then particle filters can approximate the posterior distribution of $p(x_T|o_{t=1:T})$ by maintaining a belief over the state space with a set of particles and their corresponding weights. The belief $b(x_t)$ is given by $\langle x_t^k, w_t^k \rangle_{k=1:K}$ where $\sum_{k=1:K} w_t^k = 1$.

At every time step $t$, each particle transitions to a new state $x_{t+1}^k$ by sampling from $p(x_{t+1}^k|x_t^k)$. We then receive an observation $o_{t+1}$ and the conditional likelihood of $o_{t+1}$ is multiplied with the weight $w_t^k$ before normalizing the weights *i.e.* $w_{t+1}^k = \eta \cdot p(o_{t+1}|x_{t+1}^k) \cdot w_t^k$, where $\eta$ is the normalizing factor. Usually, the belief update is followed by a resampling step where a new set of particles are sampled from the distribution of $\langle w_{t+1}^k \rangle$ and the weights are reset to $1/K$. This step handles the problem of particle degeneracy *i.e.* when most weights are nearly zero. It helps in concentrating on most promising particles. Refer to the Appendix for detailed explanation.

**Belief of graph colorings:** The task of estimating the expected coloring of a graph $\mathcal{G}$ in Eqn. 2 highly resembles the problem of state estimation in particle filtering algorithm. If we consider a graph coloring as a state, then sampling a vertex from a distribution, individualizing it and refining the graph coloring can be seen as a stochastic transition from one state to the next. We do not receive a separate observation after every transition. Instead, we use a function of the current graph embeddings to up-weight or down-weight each particle in order to approximate the pruning of non-promising subtrees. The function is akin to unnormalized observation likelihood $p(o_{t+1}|x_{t+1}^k)$ in particle filters.

Specifically, we approximate the expectation in Eqn. 2 with a set of $k$ particles and the associated set of weights which form a belief distribution over the set of particles.

$$b_t(\mathcal{G}) \approx \langle (\mathcal{G}, \boldsymbol{\pi}_t^k), w_t^k \rangle_{k=1:K}, \tag{3}$$

The belief $b_t(\mathcal{G})$ at time step $t$ approximates the posterior distribution over the set of colorings of graph $\mathcal{G}$. At each $t$, we update the belief with a stochastic transition of colorings, and the weights with an observation function. We further do resampling to focus on and expand the most promising colorings. Finally, we compute the mean coloring from the belief which gives an estimate of Eqn. 2. Note that each step can be computed in parallel for all particles and the whole process is discriminatively trained end-to-end. We call our model Particle Filter - Graph Neural Network (PF-GNN) and give the pseudo-code in Algorithm 1. Next, we describe the main steps of the algorithm in detail.

**Initialization:** We first initialize the node embeddings with node attributes or with a constant value when attributes are not present. Then, we run a 1-WL equivalent GNN for a fixed number of iterations to get the initial coloring/embeddings of the graph. Next, we instantiate our belief with $k$ copies of the embeddings along with uniformly distributed weights *i.e* $\forall k, w_1^k = 1/K$. Our belief after initialization is then $\langle (\mathcal{G}, \boldsymbol{H}_1^k), w_1^k \rangle_{k=1:K}$, where $\boldsymbol{H}_1^k$ represents the $n \times d$ matrix of node embeddings of the $k^{th}$ particle at time step 1.

**Transition step:** In the transition step, for each particle, we pick a vertex, individualize it and refine the coloring. For this, we learn a *policy function* $P(\mathcal{V}|\boldsymbol{H}_t^k; \theta)$ which is a discrete distribution on the set of vertices conditioned on the node embeddings $\boldsymbol{H}_t^k$. This can be modeled either as a separate GNN or an MLP on node embeddings, which gives a non-negative score for each vertex. We can then normalize the scores and sample a vertex from the distribution. Next, we individualize the sampled vertex by transforming its embeddings with another parameterized MLP. In this way, recoloring of vertices is also learnt from the data. Now that the symmetry in the recolored graph is broken, we refine the embeddings $\boldsymbol{H}_t^k$ with a GNN for a fixed number of iterations. This is repeated for every particle to get new set of particles $\langle \boldsymbol{H}_{t+1}^k \rangle_{k=1:K}$. Specifically, the transition step involves:

$$v \sim P(\mathcal{V}|\boldsymbol{H}_t^k; \theta) \tag{4}$$

$$\boldsymbol{M}_t^k = \mathbf{1}\mathbf{1}^\top; \quad \boldsymbol{M}_{t\ v,:}^k = MLP_{trans}(h_{v_t}^k); \quad \boldsymbol{H}_t^k = \boldsymbol{H}_t^k \odot \boldsymbol{M}_t^k; \tag{5}$$

$$\boldsymbol{H}_{t+1}^k = GNN_t(\boldsymbol{H}_t^k) \tag{6}$$

where $\boldsymbol{M}_t^k$ is a mask matrix of ones. Since we are operating on a set of $k$ particles, we need to share GNN parameters across the $k$ transitions. However, we are free to use separate GNN parameters for each step $t$. Also, it means increasing the number of particles does not lead to increase in parameters.

**Particle weights update step:** In particle filtering, an observation is received after every transition. Then an observation function is applied to estimate the likelihood of the observation, which will be used to update the particle weights. In our case, since the new information (or observation) is contained in the refinement, we model the observation as a latent variable conditioned on the refined coloring. A learnable observation function $f_{obs}(\boldsymbol{H}_{t+1}^k; \theta_o)$ is used to measure the value of the refined embeddings. Specifically, $f_{obs}(\boldsymbol{H}_{t+1}^k; \theta_o)$ is a set function approximator which outputs a non-negative score on the set of node embeddings. The output of $f_{obs}$ is then multiplied with the weights $w_t^k$, and the set of $k$ weights are normalized to get the updated distribution over $k$ colorings.

$$w_{t+1}^k = \frac{f_{obs}(\boldsymbol{H}_{t+1}^k; \theta_o) \cdot w_t^k}{\sum_k f_{obs}(\boldsymbol{H}_{t+1}^k; \theta_o) \cdot w_t^k} \tag{7}$$

Note that in the absence of intermediate observations, following Ma et al. (2020), we have used a discriminative function $f_{obs}(\boldsymbol{H}_{t+1}^k; \theta_o)$ to up/down-weight the state. Since it is discriminatively trained to optimize the end task, it performs the same function as $p(o|s)$ in particle filters.

**Resampling step:** A crucial step in the particle filter algorithm is the resampling operation. The weights of most particles tend to degenerate (*i.e.*, become near zero). To overcome this, we need to resample $K$ new particles from the discrete distribution of $\langle w_{t+1}^k \rangle_{k=1:K}$ and set their weights to $1/K$. Since this is a non-differentiable operation, we adopt the differentiable soft-resampling strategy proposed in Karkus et al. (2018). Specifically, we sample new particles from the convex combination of particle weights and uniform distribution *i.e.,* $q_t(k) = \alpha w_t^k + (1 - \alpha)1/K$, where $\alpha \in [0, 1]$ is a tunable parameter. The new weights can then be computed by importance weighting, $w_t^{\prime k} = \frac{p_t(k)}{q_t(k)} = \frac{w_t^k}{\alpha w_t^k + (1-\alpha)1/K}$, where $p_t(k)$ is the weights distribution. This is a differentiable approximation which provides non-zero gradients for the full particle chain with trade off between desired sampling distribution ($\alpha = 1$) and the uniform sampling distribution ($\alpha = 0$).

**Readout:** After $T$ iterations of individualization-refinement over the set of $k$ particles, we have the belief $\langle (\mathcal{G}, \boldsymbol{H}_T^k), w_T^k \rangle_{k=1:K}$. We can then adopt sufficiently expressive function over the belief as it is a probability distribution on the set of embeddings. Ma et al. (2019) proposed Moment Generating Function (MGF) features to readout. However, we found that they are numerically unstable in our experiments. Hence instead, we take the mean particle $\sum_{k=1:K} w_T^k \boldsymbol{H}_T^k$ for all $t \in \{1, \ldots, T\}$ and concatenate the embeddings before readout for the final prediction.

**Training:** We introduced randomness in two parts of our algorithm, one where we sample for individualization, and the other during resampling of weights. Since our resampling step is differentiable, we only need to optimize the expected loss under $P(\mathcal{I})$, which gives us a REINFORCE type gradient update along with task-specific loss. Let $\tilde{y}$ be the prediction, $y$ the target and $\mathcal{I}$ be the sequence of vertices individualized, then the expected loss over $K$ such sequences is,

$$Loss(\mathcal{G}, y) = \sum_{\mathcal{I}} P(\mathcal{I}|\mathcal{G}, \pi_{t=1:T}^{\mathcal{I}}; \theta) L(\tilde{y}(\pi_{t=1:T}^{\mathcal{I}}), y; ; \theta) \tag{8}$$

and the gradient of the expected loss is

$$\nabla Loss(\mathcal{G}, y) = \sum_{\mathcal{I}} \nabla L(\tilde{y}(\pi_{i=1:T}^{\mathcal{I}}), y; \theta) + \sum_{\mathcal{I}} \left( \nabla \log P(\mathcal{I}|\mathcal{G}, \pi_{i=1:T}^{\mathcal{I}}; \theta) \right) L(\tilde{y}(\pi_{i=1:T}^{\mathcal{I}}), y; \theta) \tag{9}$$

**On the number of paths needed to get good embeddings with less variance:** Since PF-GNN is a sampling based approach, we would want a small number of paths ($K$) to represent the search-tree with less variance. We now discuss how PF-GNN can generate good embeddings with small $K$. Firstly, $K$ can be small whenever most colorings are repeated at each level of the tree due to the presence of automorphisms in the graph. For example, in Figure 2, all the leaves are rotated versions of the same coloring. Exact isomorphism solvers find automorphisms as well as less promising subtrees on the fly and prune the search-tree. This pruning is approximated in PF-GNN by re-weighting with observation function followed by resampling. Furthermore, our individualization function is learnt end-to-end and will adapt to reduce variance if it helps the end performance. Finally, we can tune $T$ (which may increase variance) and $K$ (which reduces variance) to fit the dataset for optimal performance.

## 5 EXPERIMENTS

We evaluate PF-GNN on a set of three separate experiments *i.e.* graph isomorphism test, graph property detection and on real world datasets. Our experiments are set up to first evaluate the expressivity of PF-GNN and then to compare with other leading GNN models on real world datasets.

### 5.1 GRAPH ISOMORPHISM DETECTION

Table 1: Accuracy on graph isomorphism-test for SR25 and EXP datasets after training 100 epochs.

| DATASET | MODEL | CHEBNET | PPGN (3-WL) | GNNML3 | GCN | GAT | GIN | GNNML1 |
|---------|-------|---------|-------------|--------|-----|-----|-----|--------|
| SR25 | BACKBONE | 0.0±0.0 | 0.0±0.0 | 0.0±0.0 | 0.0±0.0 | 0.0±0.0 | 0.0±0.0 | 0.0±0.0 |
|  | +PF-GNN | - | - | - | **100.0**±0.0 | **100.0**±0.0 | **100.0**±0.0 | **100.0**±0.0 |
| EXP | BACKBONE | 0.0±0.0 | 0.0±0.0 | 100.0±0.0 | 0.0±0.0 | 0.0±0.0 | 0.0±0.0 | 0.0±0.0 |
|  | +PF-GNN | - | - | - | **100.0**±0.0 | **100.0**±0.0 | **100.0**±0.0 | **100.0**±0.0 |

Table 2: Graph classification accuracy on the CSL dataset. * indicates the backbone model.

|  | GCN | GAT | GIN* | RING-GNN | RP-GIN | 3-WL GNN | PF-GNN |
|--------|-----|-----|------|----------|--------|----------|--------|
| MEAN | 10 | 10 | 10 | 10 | 37.6 | 97.8 | **100.0** |
| MEDIAN | 10 | 10 | 10 | 10 | 43.3 | - | **100.0** |
| MAX | 10 | 10 | 10 | 10 | 53.3 | **100.0** | **100.0** |
| MIN | 10 | 10 | 10 | 10 | 10 | 30 | **100.0** |
| STD | 0 | 0 | 0 | 0 | 12.9 | 10.9 | 0 |

We first evaluate the expressiveness of PF-GNN on three graph isomorphism test datasets. 1) SR25 (Mckay, 2021). The task is to distinguish all 105 pairs of non-isomorphic strongly regular graphs, which are one of the hard class of graphs for the graph isomorphism problem. 2) EXP (Abboud et al., 2020) consists of 600 pairs of non-isomorphic graphs. For both SR25 and EXP, we train the models with all graphs and test whether the models can learn embeddings that distinguish each pair. Following Balcilar et al. (2021), we use 3-layer models for all the backbones. We run experiments with 10 different seed and report the average accuracy and standard deviation. 3) CSL (Murphy et al., 2019) consists of 150 4-regular graphs divided evenly into 10 isomorphism classes. The task is to classify the graphs into isomorphism classes that can only be accomplished by models with higher expressive power than 1-WL. As in Dwivedi et al. (2020); Murphy et al. (2019), we follow 5-fold cross validation for evaluation.

PF-GNN is a method that can be easily applied with any GNN as backbone. In Tab. 1, we compare the isomorphism detection results for models with and without PF-GNN, including GCN (Kipf & Welling, 2016), GAT (Veličković et al., 2017), GIN (Xu et al., 2018) and GNNML1 (Balcilar et al., 2021). These models are known to be at most 1-WL, and are expected to fail in the task. After augmenting these models with PF-GNN, most of them can achieve 100% accuracy. We further evaluate our method against several strong and more expressive models such as PPGN (Maron et al., 2019), GNNML3 (Balcilar et al., 2021) in Tab 1 and Ring-GNN (Chen et al., 2019b), RP-GNN (Murphy et al., 2019) and 3-WL GNN (Dwivedi et al., 2020) in Tab. 2 for the CSL dataset. As shown in Tab. 1 and Tab. 2, our proposed method consistently outperforms them experimentally. Note that PPGN is equivalent to 3-WL in expressive power. Interestingly, PF-GNN outperforms both PPGN as well as recently proposed GNNML3 on SR25 dataset. This shows that PF-GNN is able to distinguish graphs which even 3-WL equivalent models are not able to distinguish. Note that our setting differs from Balcilar et al. (2021) where there is no learning and PPGN gets 100% accuracy for EXP dataset.

### 5.2 GRAPH PROPERTIES:

Table 3: Classification accuracy on LCC and TRIANGLES datasets.

| DATASET | TEST SET | GIN | GAT | CHEBNET | RNI 12.5% | RNI 50% | RNI 87.5% | RNI | PF-GNN |
|---------|----------|-----|-----|---------|-----------|---------|-----------|-----|--------|
| LCC | N | 50 | 50 | 50 | 83 | 83 | 82 | 82 | **100** |
|  | X(LARGE) | 50 | 50 | 50 | 86 | 86 | 90 | 89 | **100** |
| TRIANGLES | ORIG | 47 | 50 | 64 | 49 | 52 | 54 | 59 | **99** |
|  | LARGE | 18 | 25 | 24 | 27 | 27 | 29 | 31 | **72** |

In this experiment, we evaluate PF-GNN on two property detection tasks where 1-WL GNNs fail. One is counting the number of triangles in TRIANGLES dataset (Knyazev et al., 2019) and the other is finding the node clustering co-efficient in LCC dataset (Sato et al., 2020). TRIANGLES (Knyazev et al., 2019) is a

Table 4: Graph regression performance on ZINC 10K and ALCHEMY 10K datasets.

| DATASET | EDGE-FEAT | METRIC | GINE-$\epsilon$ | 2-WL-GNN | $\delta$-2-GNN | $\delta$-2-LGNN | PNA | DGN | PF-GNN |
|---|---|---|---|---|---|---|---|---|---|
| ZINC | YES | MAE | $0.278_{\pm0.02}$ | $0.399_{\pm0.01}$ | $0.374_{\pm0.02}$ | $0.306_{\pm0.04}$ | $0.187_{\pm0.01}$ | $0.168_{\pm0.01}$ | $\mathbf{0.122}_{\pm0.01}$ |
| | NO | MAE | - | - | - | - | $0.320_{\pm0.03}$ | $0.219_{\pm0.01}$ | $\mathbf{0.196}_{\pm0.01}$ |
| ALCHEMY | YES | MAE | $0.185_{\pm0.01}$ | $0.149_{\pm0.01}$ | $0.118_{\pm0.01}$ | $0.122_{\pm0.01}$ | $0.162_{\pm0.01}$ | - | $\mathbf{0.111}_{\pm0.01}$ |
| | YES | LOGMAE | $-1.864_{\pm0.07}$ | $-2.609_{\pm0.03}$ | $-2.679_{\pm0.04}$ | $-2.573_{\pm0.08}$ | $-2.033_{\pm0.06}$ | - | $\mathbf{-2.96}_{\pm0.02}$ |

Table 6: Error ratio on chemical accuracy on 12 QM9 targets without using node position features.

| Acc | mu | alpha | HOMO | LUMO | gap | R2 | ZPVE | U0 | U | H | G | Cv |
|---|---|---|---|---|---|---|---|---|---|---|---|---|
| R-GCN | 3.21 | 4.22 | 1.45 | 1.62 | 2.42 | 16.38 | 17.40 | 7.82 | 8.24 | 9.05 | 7.00 | 3.93 |
| R-GIN | 2.64 | 4.67 | 1.42 | 1.50 | 2.27 | 15.63 | 12.93 | 5.88 | 18.71 | 5.62 | 5.38 | 3.53 |
| GNN-FiLM | **2.38** | 3.75 | **1.22** | **1.30** | **1.96** | 15.59 | 11.00 | 5.43 | 5.95 | 5.59 | 5.17 | 3.46 |
| PF-GNN | 3.65 | **2.06** | 1.57 | 1.52 | 2.20 | **12.76** | **3.08** | **0.9748** | **1.035** | **1.03** | **0.9747** | **1.76** |

Table 7: Graph classification on ogbg-molhiv.

| Method | ROC-AUC |
|---|---|
| GIN | $75.58_{\pm1.40}$ |
| GCN | $76.06_{\pm0.97}$ |
| PNA | $79.05_{\pm1.32}$ |
| DGN | $79.70_{\pm0.97}$ |
| Directional GSN | $\mathbf{80.39}_{\pm0.90}$ |
| PF-GNN | $80.15_{\pm0.68}$ |

large synthetic dataset with 45000 graphs. The task is to count the number of triangles in each graph. LCC dataset (Sato et al., 2020) has graphs with nodes labeled with their local clustering co-efficient. TRIANGLES has 10 graph-level classes and LCC has 3 classes at node-level. Both datasets have two different test sets. The test sets Large and X for TRIANGLES and LCC respectively come with graphs with nearly 100 nodes whereas the training set has less than 25 nodes. Better prediction in these test sets suggests that the model is able to generalize better.

Our baselines are 1-WL GNNs, Chebnet and RNI (Sato et al., 2020; Abboud et al., 2020). RNI is 1-WL GNN but with randomly initialized node features which makes it a universal approximator on graphs. We also compare with RNI on partial levels of individualization as in Abboud et al. (2020). The empirical results in Tab. 3 show that PF-GNN outperforms all RNI models as well as more expressive Chebnet on both datasets. The difference is more evident in TRIANGLES-Large where the baselines are not able to generalise to graphs with more nodes. Comparison with RNI suggests that learning the subset of nodes to individualize is helpful for generalization towards the end-task.

## 5.3 REAL WORLD BENCHMARKS

We now evaluate PF-GNN on various real-world benchmark datasets including ZINC (Jin et al., 2018; Dwivedi et al., 2020), ALCHEMY (Chen et al., 2019a) and QM9 (Ruddigkeit et al., 2012; Ramakrishnan et al., 2014) and Ogbg-molhiv (Hu et al., 2020). For ZINC and ALCHEMY, we use the fixed training set of 10K graphs and 1K graphs each for testing and validation. QM9 dataset consists of nearly 130000 graphs. As is the standard practice, we use the 10K each for testing and validation, and rest of the graphs for training. For ogbg-molhiv (Hu et al., 2020), we use full 9-dimensional node features and adopt the standard scaffold split. We report mean absolute error for the former three datasets, and ROC-AUC for ogbg-molhiv. Details are in Appendix A.5.5.

Table 5: Mean absolute error on jointly trained QM9 targets w/ position coordinates.

| Data set | QM9 |
|---|---|
| GINE-$\epsilon$ | $0.081_{\pm0.003}$ |
| MPNN | $0.034_{\pm0.001}$ |
| 1-2-GNN | $0.068_{\pm0.001}$ |
| 1-3-GNN | $0.088_{\pm0.007}$ |
| 1-2-3-GNN | $0.062_{\pm0.001}$ |
| 3-IGN | $0.046_{\pm0.001}$ |
| $\delta$-2-LGNN | $0.029_{\pm0.001}$ |
| Dimenet | $0.019_{\pm0.001}$ |
| PF-GNN | $\mathbf{0.017}_{\pm0.001}$ |

For ZINC, we compare with neural $k$-WL and their sparse versions (Morris et al., 2020) along with other recent models including PNA (Corso et al., 2020), DGN (Beani et al., 2021). For Alchemy, we compare with sparse $k$-GNNs. Scores of PNA on Alchemy are reported by running their open source code. Tab. 4 shows consistent improvement of error rates in both Zinc and Alchemy datasets. In both the datasets, PF-GNN shows strong performance in comparison to all reported models.

For QM9, we compare with most of the strong baselines including sparse $k$-GNN, 3-IGN (Maron et al., 2018) and Dimenet (Klicpera et al., 2020). In Tab. 5, we report MAE scores averaged over all 12 targets. PF-GNN achieves best performance compared to all the baseline models. Note that in this setting all targets are jointly trained and include position co-ordinates used either as node or edge features. It has been observed that QM9 results are strongly correlated with positional features (Klicpera et al., 2020). This is not an appropriate setting to evaluate PF-GNN, as the main motivation of PF-GNN is to identify nodes without distinguishing features. Therefore, we also report

Table 8: Ablation results of PF-GNN with different $T$ and $K$. Number on the left are for TRIANGLES-large, on the right are for CSL.

| Acc | K=1 | K=2 | K=4 | K=8 | K=16 |
|---|---|---|---|---|---|
| T=1 | 32 \| 60 | 29 \| 60 | 35 \| 60 | 38 \| 60 | 40 \| 50 |
| T=2 | 47 \| 96 | 40 \| 93 | 47 \| 97 | 42 \| 90 | 47 \| 92 |
| T=3 | 44 \| 94 | 59 \| 95 | 59 \| 98 | 67 \| 95 | **69** \| 97 |
| T=4 | 42 \| 89 | 43 \| 93 | 47 \| 88 | 46 \| **100** | 55 \| 95 |
| T=8 | 33 \| 92 | 33 \| 72 | 33 \| 82 | 31 \| 85 | 45 \| 98 |

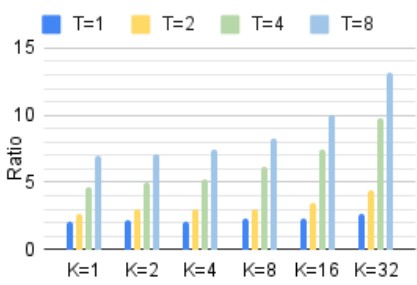

Figure 3: Runtime ratio of PF-GNN on SR25 dataset.

results in Tab. 6 for the setting where no positional features are used. We compare with the baseline results reported in Brockschmidt (2020). PF-GNN outperforms GNN-FiLM (Brockschmidt, 2020) in most of the targets. For Ogbg-molhiv, we compare with some of the strong baselines including GCN, GIN, PNA, DGN and GSN (Bouritsas et al., 2020). In Tab. 7, we report ROC-AUC scores which shows PF-GNN achieves competitive performance against the baselines.

## 5.4 ABLATION STUDIES

**Performance:** We first run ablative experiments of PF-GNN with different number of IR steps ($T$) and number of particles ($K$). As presented in Tab. 8, best performance is achieved for TRIANGLES-large at $T = 3$ and $K = 16$, and for CSL at $T = 4$ and $K = 8$. Note that a single IR layer is insufficient to reach the best performance on both TRIANGLES-large and CSL datasets. Subsequent addition of IR steps is beneficial up to a few rounds but negatively impacts the performance on the end task after $T = 4$ for these two tasks. This is not the case for the number of particles $K$. With higher $K$, the score either improves or remains the same. This empirically verifies that higher number of samples help in better approximation of the true expectation of the embeddings.

Table 9: Effect of resampling and policy loss. Results on TRIANGLES dataset.

| | Train | Orig | Large |
|---|---|---|---|
| PF-GNN | 99 | 99 | 72 |
| w/o resampling | 99 | 99 | 63 |
| w/o policy loss | 97 | 96 | 46 |

We further study two ablative models of PF-GNN, 1) without resampling updates and 2) training without the policy gradient loss. Tab. 9 shows that the test performance of PF-GNN on TRIANGLES-large dataset depends on both the resampling operation as well as training with the policy loss. Interestingly, the policy loss seems to help significantly which suggests that learning the nodes to individualize is important. Furthermore, TRIANGLES-large test set contains graphs with larger number of nodes as compared to the graphs in training data. Hence, the results indicate that both operations can help our model to generalize better. Additional ablation experiments are included in the Appendix A.5

**Runtime:** PF-GNN is efficient in terms of its runtime given its expressive power. Since, we run a GNN in each step for fixed iterations, the runtime of PF-GNN is bounded by $\mathcal{O}(nT)$ depth $T$ with parallel computation of $K$ paths. We further empirically evaluate the increase in runtime of PF-GNN compared to the 1-WL GNN model. Fig. 3 plots the ratio of increase in runtime compared to the base GNN, as we increase number of particles and depth. The results confirm that the runtime increase is only linear as we keep increasing $T$. Note that we do not need longer depth as a few individualizations are sufficient to break most of the symmetry in graphs. Additional runtime results in Appendix A.5.3.

## 6 CONCLUSION

Message passing GNNs are known to be not fully expressive for learning functions on graph structured data. In this paper, we first define a way of learning universal representations of graphs by guiding the learning process with exact isomorphism solver techniques. However, learning these representations can take exponential time in worst case. To be practically useful, we proposed PF-GNN, a differentiable particle filtering based model to approximately learn the universal representations. PF-GNN learns highly discriminative representations while being efficient in runtime requirements. It is flexible and can be used with any GNN as a backbone. Experimental results on various benchmarks demonstrate that PF-GNN can distinguish even those graphs not distinguishable by 3-WL equivalent models and gives competitive results on real-world datasets.

## ETHICS STATEMENT

Our work is a generic method to improve the expressive power of any GNN backbone models. As our work applies to GNNs generally, its ethical impacts (both positive and negative) will be on the broad approach of GNNs rather than on any specific application.

## REPRODUCIBILITY STATEMENT

The complete proofs for Theorem 1 and Theorem 2 are included in Appendix A.3. For experiments, the settings are stated in Section 5, and more details of the model architectures and hyperparameter for training are specified in Appendix A.5.5. Details of the datasets used can be found in Appendix A.5.7. To further facilitate reproducibility of the work, we provide publicly available code at https://github.com/pfgnn/PF-GNN.

## ACKNOWLEDGEMENT

This research is supported in part by the National Research Foundation, Singapore under its AI Singapore Program (AISG Award No: AISG2-RP-2020-016).

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

# A    APPENDIX

## A.1    PARTICLE FILTER ALGORITHM

A Particle filter estimates the posterior distribution of the states of a Markov process when noisy/partial observations are received. It uses a set of samples or particles to represent the posterior distribution of the state given a series of observations. One of the main benefit of particle filtering is that the particle set can approximate any arbitrary distributions,

Consider a Markov process with the state transition probability given by $p(x_t|x_{t-1})$ where $x_t$ is the state at time step $t$. For every transition, we receive an observation $o_t$ described by observation probability of $p(o_t|x_t)$. Given a series of observations $o_{t=1:T} = \{o_1, \ldots, o_T\}$, particle filters estimate the posterior probability $p(x_T|o_{t=1:T})$, of the state $x_T$ conditioned on the sequence of observations.

Specifically, particle filters sequentially approximate the posterior distribution by maintaining a belief over the states $b(x)$ with a set of $K$ *particles*, *i.e.*, weighted samples from the probability distribution,

$$b_t(x) \approx \langle x_t^k, w_t^k \rangle_{k=1:K}, \tag{10}$$

where $\sum_k w_t^k = 1$, $K$ is the number of particles, $w_t^k$ is the particle weight and $x_t^k$ is the particle state at time step $t$.

The particles are sequentially updated with three steps: transition step, observation update step and the resampling step.

**Transition step:**

At each time step, the particles transition to new states depending on the current state. For each of the $K$ particles, the new state is computed by sampling from the probabilistic transition model given by,

$$x_t^k \sim p(x_t|x_{t-1}^k), \tag{11}$$

where $p(x_t|x_{t-1}^k)$ is the probability of state $x_t$ given the previous state $x_{t-1}^k$.

**Observation update step:**

After a transition step, we receive an observation which is a noisy signal for the state of the system. Therefore, after every transition, the weights of the belief need to be updated with the observation received. The observation compatibility function given by $f_{obs}(o_t|x_t)$, measures the conditional likelihood of the observation given the state. Since, we have $K$ particles, we measure the observation likelihood for each one of them by computing

$$f_{o_t}^{\ k} = f_{obs}(o_t|x_t^k) \tag{12}$$

The weights of the particle set are updated by mulitplying with observation model and then normalizing the weights across the $K$-particle set.

$$\tilde{w}_t^k = \eta^{-1} f_{o_t}^{\ k} w_{t-1}^k, \tag{13}$$

where $\eta = \sum_{j=1:K} f_t^j w_{t-1}^j$ is the normalization factor.

**Resampling step:**

A very common issue in sequential state estimation with a set of particles is the problem of particle degeneracy when most of the weight tends to be concentrated in one or two particles. This issue is resolved by *resampling* a set of new particles from the weights distribution. The new set of particles are sampled from the weighted distribution $\langle \tilde{w}_t^k \rangle_{k=1:K}$ with repetition. Following the resampling operation, the weights are reset to a uniform distribution by setting,

$$w_t^k = 1/K. \tag{14}$$

The new belief represented by the particle set $\langle w_t^k, x_t^k \rangle$ approximates the same distribution, but devotes its space to the most promising particles.

At any step $t$, the state can be estimated by taking the weighted mean of the particle set,

$$\overline{x}_t = \sum_k w_t^k x_t^k \tag{15}$$

A.2 DERIVATION OF THE GRADIENT UPDATE

The prediction $\tilde{y}_T$ generated by the PF-GNN depends on multiple sampling steps. Therefore, in order to optimize the parameters, we minimize the expected loss under the random variables sampled during the process. The two random variables in PF-GNN are the particles selected during resampling step with weights $\langle w_t^k \rangle_{k=1:K}$ and the sequence of individualization vertices $\mathcal{I}$. Since, we use soft-resampling and are able to backpropagate through the weights $\langle w_t^k \rangle_{k=1:K}$, we only optimize the expected loss with random variable $\mathcal{I}$.

Let $y$ be the prediction target, $\mathcal{I}$ be the sequence of vertices individualized in one particle chain from root to leaf, $\pi_{i=1:T}^{\mathcal{I}}$ be the colorings generated in one sequence of individualizations $\mathcal{I}$. Then the expected loss is over all such sequences.

$$Loss(\mathcal{G}, y) = \sum_{\mathcal{I}} P(\mathcal{I}|\mathcal{G}, \pi_{t=1:T}^{\mathcal{I}}; \theta) L(\tilde{y}(\pi_{t=1:T}^{\mathcal{I}}), y; \theta) \tag{16}$$

where $P(\mathcal{I}|\mathcal{G}, \pi_{t=1:T}^{\mathcal{I}}; \theta)$ is the probablity of the full chain for $t = 1 : T$ of the sequence $\mathcal{I}$ and $L(\tilde{y}(\pi_{t=1:T}^{\mathcal{I}}), y; \theta)$ is the loss of the final model prediction $\tilde{y}$ and the target $y$. Note that the both $P(\mathcal{I}|\cdot; \theta)$ and $L(\tilde{y}, y; \theta)$ share some of parameters. Consequently, we cannot separately learn both the functions. Taking the gradient of the loss, we have:

$$\nabla Loss(\mathcal{G}, y) = \nabla \sum_{\mathcal{I}} P(\mathcal{I}|\mathcal{G}, \pi_{t=1:T}^{\mathcal{I}}; \theta) L(\tilde{y}(\pi_{t=1:T}^{\mathcal{I}}), y; \theta) \tag{17}$$

$$= \sum_{\mathcal{I}} \nabla \Big( P(\mathcal{I}|\mathcal{G}, \pi_{t=1:T}^{\mathcal{I}}; \theta) L(\tilde{y}(\pi_{t=1:T}^{\mathcal{I}}), y; \theta) \Big) \tag{18}$$

$$= \sum_{\mathcal{I}} P(\mathcal{I}|\mathcal{G}, \pi_{t=1:T}^{\mathcal{I}}; \theta) \nabla L(\tilde{y}(\pi_{t=1:T}^{\mathcal{I}}), y; \theta)$$
$$+ \nabla P(\mathcal{I}|\mathcal{G}, \pi_{t=1:T}^{\mathcal{I}}; \theta) L(\tilde{y}(\pi_{t=1:T}^{\mathcal{I}}), y; \theta) \tag{19}$$

$$= \sum_{\mathcal{I}} P(\mathcal{I}|\mathcal{G}, \pi_{t=1:T}^{\mathcal{I}}; \theta) \nabla L(\tilde{y}(\pi_{t=1:T}^{\mathcal{I}}), y; \theta)$$
$$+ \sum_{\mathcal{I}} P(\mathcal{I}|\mathcal{G}, \pi_{t=1:T}^{\mathcal{I}}; \theta) \nabla \log P(\mathcal{I}|\mathcal{G}, \pi_{t=1:T}^{\mathcal{I}}; \theta) L(\tilde{y}(\pi_{t=1:T}^{\mathcal{I}}), y; \theta) \tag{20}$$

In the second step, we used the product rule of differentiation and in last step, we have used the identity $\nabla p(\mathcal{I}|\cdot) = p(\mathcal{I}|\cdot) \cdot \nabla \log p(\mathcal{I}|\cdot)$. This gives us two additive expectation terms under the distribution of $P(\mathcal{I}|\mathcal{G}, \pi_{t=1:T}^{\mathcal{I}}; \theta)$ to optimize in the loss function.

$$\nabla Loss(\mathcal{G}, y) = \mathbb{E}_{\mathcal{I} \sim P(\mathcal{I}|\cdot)} \nabla L(\tilde{y}(\pi_{t=1:T}^{\mathcal{I}}), y; \theta)$$
$$+ \mathbb{E}_{\mathcal{I} \sim P(\mathcal{I}|\cdot)} \big( \nabla \log P(\mathcal{I}|\mathcal{G}, \pi_{t=1:T}^{\mathcal{I}}; \theta) \big) L(\tilde{y}(\pi_{t=1:T}^{\mathcal{I}}), y; \theta) \tag{21}$$

We optimize the expectations by the set of $K$ samples of $\mathcal{I}$ drawn from the distribution of $P(\mathcal{I}|\mathcal{G}, \pi_{t=1:T}^{\mathcal{I}}; \theta)$.

A.3 PROOF FOR THEOREMS

**Theorem 1.** Consider a $n$-vertex graph $\mathcal{G}$ with no distinct node attributes. Assume we use universal multiset function approximators for target-cell selection and graph pooling function $\psi$, and a GNN with 1-WL equivalent expressive power for color refinement, then the representation $f(\mathcal{G})$ in Eqn. 1 is permutation-invariant and can uniquely identify $\mathcal{G}$.

*Proof.* First, note that the set of leaves generated by the search tree of the exact graph-isomorphism solvers for a given $n$-vertex graph respects permutation-equivariance and contains the canonical representation as the largest element, hence uniquely identifies the graph. Please refer McKay & Piperno (2014) for a detailed proof. To complete the proof, observe that with sufficiently expressive networks, we can approximate the target-cell selector and individualization algorithms in the exact solver arbitrarily closely. Using 1-WL equivalent GNN, we can also approximate the refinement operation arbitrarily closely. Repeated applying these operations give an equivalent set of leaves as

those produced by the exact solver. Finally, using a universal multiset approximator $\psi$ in conjunction with an MLP $\rho$ which is also universal approximator allows us to distinguish distinct sets of leaves from non-isomorphic graphs. $\qquad\square$

**Theorem 2.** *Assume we need $D$ dimensional embeddings for some $D > 0$, in order to uniquely represent any graph of $n$-vertices. Consider two $n$-vertex graphs $\mathcal{G}_1$ and $\mathcal{G}_2$ whose universal representations in Eqn. 2 are $f_1$ and $f_2$ after $T$ levels of the search tree. Let the max-norm distance between $f_1$ and $f_2$ be at least $d$ i.e. generating the full level-$T$ search tree for both graphs will separate them by $d$. Assume that the values in embeddings generated by the tree are strictly bounded in the range of $[-M, M]$. Then, with probability of at least $1 - \delta$, the approximate embeddings, generated by sampling $K$ paths from root to leaves, are separated in max-norm distance by $d \pm \epsilon$ provided the number of root to leaf paths sampled are at least*

$$K \in \mathcal{O}\left(\frac{M^2 \ln(\frac{4D}{\delta})}{\epsilon^2}\right)$$

*Proof.* Observe that we independently sample $K$ paths from root to leaf of the search tree. After $T$ steps in the tree, if the mean of the embeddings is $f^T(\mathcal{G}) = \frac{1}{|\Pi^T|} \sum_i \psi(\mathcal{G}, \boldsymbol{\pi}_i^T)$, then we have $K$ independant samples to approximate the mean. For clarity, we omit $\mathcal{G}$ and refer actual expectation as $f^T$ and sample approximation with $\overline{f}^T$.

Note that by the Hoeffding bound (Hoeffding, 1994; Shalev-Shwartz & Ben-David, 2014) for mean of $K$ independent random variables $Y_1, \ldots, Y_K$ in range $[-M, M]$ with expected value $\mu$, we have

$$P\left(|\overline{Y} - \mu| \geq \zeta\right) \leq 2\exp\left(-\frac{K\zeta^2}{2M^2}\right) \tag{22}$$

Assume we need $D$ dimensions for some $D > 0$, in order to fully represent any $n$-vertex graph. Since, we are independently sampling from the leaves of the tree, the Hoeffding bound holds for each of the estimated value in the $D$-dimensional vector. Hence, for any $i^{th}$ value in the graph embedding,

$$P\left(|\overline{f}_{iT} - f_{iT}| \geq \zeta\right) \leq 2\exp\left(-\frac{K\zeta^2}{2M^2}\right) \tag{23}$$

Since this bound holds for every value of the embedding. we can bound the maximum error for any of the $D$ estimates with the "union bound" of probability for all the $D$ values simultaneously.

$$P\left(\max_i |\overline{f}_{iT} - f_{iT}| \geq \zeta\right) \leq 2D\exp\left(-\frac{K\zeta^2}{2M^2}\right) \tag{24}$$

This gives us the max-norm distance between the estimated and the expected embeddings. Using $\zeta = \epsilon/2$, we get

$$P\left(\|\overline{f}_{iT} - f_{iT}\| \geq \epsilon/2\right) \leq 2D\exp\left(-\frac{K\epsilon^2}{8M^2}\right) \tag{25}$$

Now, assume two non-isomorphic graphs $\mathcal{G}_1$ and $\mathcal{G}_2$, are separated in max-norm distance by $d$ i.e. $\|f_{1T} - f_{2T}\| = d$, at level $T$ of the search tree. With at least $K$ samples, we can union bound the error in distance between the two graphs by $\epsilon$.

$$P\left(\big|\, \|\overline{f}_{1T} - \overline{f}_{2T}\| - \|f_{1T} - f_{2T}\| \,\big| \geq \epsilon\right) \leq P\left(\|\overline{f}_{1T} - f_{1T}\| \geq \frac{\epsilon}{2}\right) + P\left(\|\overline{f}_{2T} - f_{2T}\| \geq \frac{\epsilon}{2}\right) \tag{26}$$

$$\leq 4D\exp\left(-\frac{K\epsilon^2}{8M^2}\right) \tag{27}$$

Bounding the failure probability with $\delta$, we get the minimum samples needed for the approximation as:

$$K \geq \frac{8M^2 \ln(\frac{4D}{\delta})}{\epsilon^2} \tag{28}$$

$$K \in \mathcal{O}\left(\frac{M^2 \ln(\frac{4D}{\delta})}{\epsilon^2}\right) \tag{29}$$

$\qquad\square$

## A.4 PF-GNN Algorithm

---
**Algorithm 1** PF-GNN Algorithm

---
1: **Input:** Graph $\mathcal{G} = (\mathcal{V}, \mathcal{E})$; GNN; $f_{ind}$; $f_{obs}$; $P(\cdot; \theta)$
2: **Output:** $\tilde{y}, b_T(\mathcal{G})$
3: Initialize $\boldsymbol{H}_0 = \boldsymbol{I}$
4: $b_1(\mathcal{G}) = \langle \boldsymbol{H}_1 = GNN(\boldsymbol{H}_0), w_1^k = 1/K \rangle_{k=1:K}$         (initial belief)
5: **for** $t = 1$ **to** $T$ **do**
6:    **for** $k = 1$ **to** $K$ **do**
7:       $v \sim P(\mathcal{V} | \boldsymbol{H}_t^k; \theta)$
8:       $\tilde{\boldsymbol{H}}_t^k = f_{\text{ind}}(\boldsymbol{H}_t^k, v)$
9:       $\boldsymbol{H}_{t+1}^k = GNN(\tilde{\boldsymbol{H}}_t^k)$         (transition update)
10:    **end for**
11:    $\langle \tilde{w}_t^k \rangle_{k=1:K} = \eta \langle f_{obs}(\boldsymbol{H}_{t+1}^k; \theta_o) \cdot w_t^k \rangle_{k=1:K}$         (observation)
12:    $\langle w_{t+1}^k \rangle_{k=1:K} \sim \langle \tilde{w}_t^k \rangle_{k=1:K}$         (resampling step)
13: **end for**
14: $b_T(\mathcal{G}) = \langle \boldsymbol{H}_T, w_T^k \rangle_{k=1:K}$         (final belief)
15: $\tilde{f}(\mathcal{G}) = CONCAT(\sum_{k=1:K} w_t^k \boldsymbol{H}_t^k)_{t=1:T}$
16: $\tilde{y} = READOUT(\tilde{f}(\mathcal{G}))$
17: **Return** $\tilde{y}, b_T(\mathcal{G})$

---

## A.5 Additional Experiments

In this subsection, we provide further experiments in order to evaluate various components of the PF-GNN algorithm.

### A.5.1 Impact of individualization function:

In PF-GNN, individualization method involves transforming the selected node embedding via an MLP. In this ablation study, we seek to learn whether learning the individualization method is useful for the end performance. For this, we replace the learnable MLP with non-learnable random features sampled from standard Normal distribution as used in Abboud et al. (2020) and compare the performance on TRIANGLES and CSL datasets. Moreover, we compare both methods of individualization for various partial levels of node-feature perturbations as partial individualizations were noted to be helpful in RNI (Abboud et al., 2020). In partial RNI, only part of embedding dimensions are perturbed. For example, 50% refers to 50% embedding dimensions of the selected node are perturbed in each IR step. PF-GNN with T=3 and K=4 was used in the experiments. We train all models with randomized individualization till convergence in this experiment.

Table 10: Effect of MLP vs Random individualization functions for various levels of partial individualization.

| Dataset | MLP 100% | MLP 87.5% | MLP 50% | MLP 12.5% | RANDOM 100% | RANDOM 87.5% | RANDOM 50% | RANDOM 12.5% |
|---|---|---|---|---|---|---|---|---|
| Triangle-Large | 68.9 | 67.9 | 68.3 | 61.4 | 64.5 | 57.4 | 49.3 | 27.5 |
| CSL | 100 | 100 | 100 | 100 | 100 | 100 | 82.6 | 48.0 |

Results in Tab. 10 suggest that learnable MLP is better than random perturbations for node individualization. For partial individualization, MLP performs similarly or better than random features. For full individualization, the difference is less clear in CSL dataset as both methods give similar result. However, very small levels of partial random perturbation seem to harm performance in both datasets.

### A.5.2 Impact of sequential individualization:

We now study the impact of sequential individualization of PF-GNN. In the ablation study of the main paper, we show in Tab. 9 that sequential individualization without learning significantly reduces performance i.e. which nodes are selected in each IR step is important. We further examine the

effects of sequential individualization compared to simultaneous individualization. For this, we compare following ablation models of PF-GNN: 1) PF-GNN with T=3 steps of IR, 2) PF-GNN-A: PF-GNN with a single IR step but 3 randomly chosen nodes individualized in the same step. 3) PF-GNN-B: single IR step but 6 random nodes are individualized at once 4) PF-GNN C: single IR step with 9 nodes individualized. Note that we individualize same number of nodes for PF-GNN method (sequential) and PF-GNN-A method (simultaneous). We train all models till convergence in this experiment.

Table 11: Effect of sequential individualization in PF-GNN.

| Dataset
#Individualized nodes | PF-GNN
3 | PF-GNN-A
3 | PF-GNN-B
6 | PF-GNN-C
9 |
|---|---|---|---|---|
| Triangles-large | 68.7 | 35.5 | 36.2 | 37.6 |
| CSL | 100 | 49.23 | 42.67 | 41.33 |

Results in Tab. 11 show that sequentially individualizing selected nodes is better than individualizing a subset of nodes at once on both the datasets. Since most nodes are indistinguishable initially, it is hard to learn the optimal subset in one step. With sequential IR, the graph-coloring is refined in each step and the model can learn to successively pick nodes which are likely to generate the best refinement. Therefore, "Sequential IR process with learning" helps in finding those subset of nodes. Furthermore in CSL dataset, performance goes down as we increase the number of nodes individualized at once, suggesting that larger unguided randomness may not be helpful. Overall, the results suggest that PF-GNN's data-driven sequential refinement improves GNN's expressivity while adding minimal randomness.

### A.5.3 Runtime analysis

In this subsection, we further study the running time behaviour of PF-GNN in contrast to the base GNN models. Assuming GNN takes linear time for bounded-degree graphs $\mathcal{O}(n)$, runtime of PF-GNN is $\mathcal{O}(nT)$ for $T$ number of IR steps. Note that the computation of $K$ paths can be parallelized. In Tab. 12, we provide additional empirical runtime results of PF-GNN models for different $K$ and $T$ values for the Zinc and Alchemy datasets. We also compare with the runtime of the standalone backbone GNN model. The numbers in brackets indicate the ratio of the runtime of PF-GNN w.r.t that of backbone GNN. For fair comparison, all models are trained on a single GPU with a batch size of 64 with PNA convolution as the base GNN. The results indicate that runtime increase is only linear in $T$.

Table 12: Runtime on ALCHEMY and ZINC datasets. Numbers in the brackets indicate the runtime ratio with respect to the base GNN.

| | GNN | K=4, T=1 | K=4, T=2 | K=4, T=3 | K=8, T=1 | K=8, T=2 | K=8, T=3 |
|---|---|---|---|---|---|---|---|
| Alchemy | 6.72s (1) | 11.70s (1.74) | 14.82s ( 2.20) | 18.24s (2.71) | 12.28s (1.82) | 17.35s ( 2.58) | 23.57s (3.51) |
| Zinc | 8.42s (1) | 13.12s (1.73) | 19.24s ( 2.54) | 26.24s (3.48) | 17.14s (2.03) | 27.11s (3.21) | 38.57s (4.47) |

Furthermore, PF-GNN is a flexible model. We can reduce the number of GNN iterations within each IR step, if we want to keep the same time as the base GNN. Even so, we can still get performance improvements with PF-GNN. Below we report results of the standalone base GNN model with 6 iterations and PF-GNN with $T = 1$ and $T = 2$ but 6 GNN message passing steps in total. Results in Tab. 13 show that PF-GNN improves upon GNN with the same budget of total GNN layers. Additionally, the results further empirically support that the improvements are a result of IR process and not merely addition of more GNN layers.

Table 13: Runtime vs MAE comparison of PF-GNN with base GNN with a total 6 GNN layers.

| Model | GNN steps | Zinc (MAE) | Zinc (time) | Alchemy (MAE) | Alchemy (time) |
|---|---|---|---|---|---|
| 6-step base GNN | [6] | 18.14 | 8.92s | 15.75 | 6.82s |
| PF-GNN T=1 | [4,2] | 13.37 | 12.33s | 11.66 | 9.06s |
| PF-GNN T=2 | [2, 2, 2] | 14.01 | 17.57s | 11.73 | 11.17s |

### A.5.4 Graph property testing

In the main paper, we studied graph properties like triangle counting and local clustering co-efficient and found the PF-GNN outperforms 1-WL equivalent GNNs. In this subsection, we study another set of fundamental graph properties that GNNs cannot capture. These properties are connectivity, bipartiteness and planarity (Kriege et al., 2018). For all three tasks, we generate synthetic datasets of 100 graphs for training and 40 graphs for testing with equal number of positive and negative samples. Each graph consists of 20 nodes each. For connectivity dataset, we first generate two random connected graphs which forms a single instance of non-connected graph and add an edge between the two to make it a positive sample of connected graph. For planar datasets, we generate random regular graphs and check for planarity. We repeat till we have equal number of planar and non-planar graphs. For bipartiteness, we generate a random bipartite graph and create a positive sample by adding an edge between two groups of vertices and negative sample by adding an edge within a group of vertices. Tab. 14 shows that GIN model fails on all three tasks while PF-GNN reaches near optimal accuracy on each one of them.

Table 14: GNN and PF-GNN on fundamental graph property testing.

|          | Connectivity | Planarity | Bipartiteness |
|----------|--------------|-----------|---------------|
| GIN      | 62.4         | 50.5      | 55.7          |
| PF-GNN   | 97.5         | 98.7      | 99.1          |

### A.5.5 Hyperparameters and architecture details

In this subsection, we report the hyperparameters used in all experiments from Section 5 in Tab. 15. The architecture of PF-GNN is specified by the backbone model, number of particles and number of IR steps. We also report the hyperparameters used for training, including batch size, number of embedding dimensions and the weight for policy loss. The learning rate is set at 1e-3 for SR25 and EXP, and we use a scheduler to reduce it based on the improvement of metrics for other datasets.

Table 15: Details of hyperparameters used for each dataset. $K$: number of particles, $T$: number of IR steps, $\gamma$: weight of policy loss.

| Dataset      | Backbone | K  | T | $\gamma$ | Batch size | #Hidden dim | #Epochs |
|--------------|----------|----|---|----------|------------|-------------|---------|
| SR25         | -        | 4  | 8 | 0.1      | 128        | 64          | 100     |
| EXP          | -        | 4  | 8 | 0.1      | 128        | 64          | 100     |
| CSL          | GIN      | 8  | 3 | 1        | 16         | 64          | 500     |
| TRIANGLES    | PNA      | 24 | 3 | 1        | 128        | 64          | 500     |
| LCC          | GIN      | 4  | 2 | 0.1      | 64         | 64          | 500     |
| ZINC         | PNA      | 8  | 3 | 0.1      | 64         | 150         | 500     |
| ALCHEMY      | PNA      | 8  | 4 | 0.1      | 64         | 150         | 500     |
| QM9          | MPNN     | 4  | 2 | 0.1      | 64         | 64          | 300     |
| Ogbg-molhiv  | PNA      | 8  | 2 | 0.1      | 256        | 120         | 200     |

**Baseline results:** The baseline results mentioned in Tab. 1 are reproduced from the open sourced code from Balcilar et al. (2021) and those in Tab. 2 are from Dwivedi et al. (2020). We reproduced the results from Knyazev et al. (2019) for Tab. 3.

For Tab. 4, we reproduced results for PNA from the open sourced code and rest of the results are reported from Morris et al. (2020) for $k$-GNNs and Beani et al. (2021) for DGN. For QM9 results in Tab. 5, we report results from Morris et al. (2020); Klicpera et al. (2020) and from Brockschmidt (2020) for Tab. 6. For Tab. 7, we get the baseline results from the OGB leaderboard (Hu et al., 2020).

### A.5.6 Additional Empirical Results

Two additional sets of experiments are done to complement our empirical evaluation. In Abboud et al. (2020), they use a different setting for EXP dataset as compared to our experiment in Tab. 1, which is 10-fold cross-validation. We follow their setting and make further comparison with more expressive baselines in Tab. 16. PF-GNN again achieved 100% accuracy.

We also include one additional ablation study on number of IR steps and number of particles. We test the performance for each architecture on the easier TRIANGLES-orig test set, where the number of nodes for each graph is similar to that of the graphs in training set. As shown in Tab. 17, models with L=1 get substantially less accuracy than the deeper ones. Furthermore, increasing both K and T helps with the performance.

Table 16: Graph isomorphism test on EXP dataset with 10-fold cross-validation. Results of the baselines are from Abboud et al. (2020).

| DATA SET | EXP |
|---|---|
| GCN-$r$ | 98.0 $\pm_{1.85}$ |
| PPGN | 50.0 |
| 1-2-3-GCN-L | 50.0 |
| PF-GNN | **100.0** $\pm_{0.00}$ |

Table 17: Ablation results of PF-GNN with different number of IR steps (T) and different number of particles (K). Results are on TRIANGLES-orig test set.

| ACC | K=1 | K=2 | K=4 | K=8 | K=16 |
|---|---|---|---|---|---|
| T=1 | 85 | 82 | 87 | 89 | 91 |
| T=2 | 96 | 93 | 94 | 96 | 96 |
| T=3 | 95 | 98 | 98 | 99 | 99 |
| T=4 | 99 | 99 | 99 | 99 | 99 |
| T=8 | 99 | 99 | 99 | 99 | 99 |

### A.5.7 DATASETS

**SR25:**

This dataset (Mckay, 2021) contains 15 strongly regular graphs, where each of them has 25 nodes and each node has 12 neighbors. 5 common neighbors are shared by connected nodes while 6 common neighbor are shared by disconnected nodes. There are in total 105 pairs of non-isomorphic graphs. The task is to distinguish all pairs. This is a challenging task because strongly regular graphs are known to be one of the hardest class of graphs for detecting isomorphism. It is to be noted that SR25 dataset is used as one of the benchmarks for isomorphism detection task.

**EXP:**

This dataset (Abboud et al., 2020) consists of 600 pairs of non-isomorphic graphs. Each graph in in EXP dataset encodes a propositional formula. Classifying the graph is equivalent to determining if the propositional formula is satisfiable or not.

For both SR25 and EXP, we train the models with all graphs and test whether the models can learn embeddings that distinguish each pair.Following, Balcilar et al. (2021) we use 3-layer models for all the backbones in Tab. 3, and 8 IR steps with 4 particles for models with PF-GNN. We run experiments with 10 different seed and report the average accuracy and standard deviation.

**CSL:**

This dataset from Murphy et al. (2019) consists of 150 4-regular graphs with edges forming a cycle and contain skip links. The 150 graphs are divided evenly into 10 isomorphism classes based on the skip length. All graphs are 1-WL equivalent. The task is to classify the graphs into isomorphism classes that can only be accomplished by models with higher expressive power than 1-WL. As in Dwivedi et al. (2020); Murphy et al. (2019), we follow 5-fold cross validation for evaluation. PF-GNN results are obtained with 3 IR steps, 8 particles and GIN as backbone.

**TRIANGLES:**

The TRIANGLES dataset, released by (Knyazev et al., 2019), consists of 45000 graphs and the task is to count the number of triangles in each graph. Interestingly, the dataset comes with two test splits; test-orig and test-large, both of which have 5000 graphs. The training set as well as the test-orig set consists of graphs with $< 25$ nodes while the test-large set has $< 100$ nodes in each graph. Better prediction in test-large suggests that the model is able to generalize better. It is a graph classification problem with 10 classes.

**LCC:**

LCC dataset (Sato et al., 2020) consists of 2000 random 3-regular graphs with 1000 graphs each for training and testing. The nodes are labelled with their local clustering coefficient (LCC) value. LCC measures how close a node's neighbours are to being a clique. It is yet another property that 1-WL GNNs fail to identify. It is a node level classification problem with 3 classes. The dataset consists of two test sets $N$ and $X$. The training set and test-$N$ set have graphs with 20 nodes and test-$X$ has graphs with 100 nodes. A good performance on LCC-X further shows the generalization ability of the model to larger graphs.

**ZINC:**

ZINC is a real-world molecular dataset consisting of more than 250K graphs. We select the popularly used subset made available by Dwivedi et al. (2020) which contains 12000 graphs. The dataset comes with standard split of 10000 graphs for training set and 1000 graphs each for testing and validation sets. It contains graphs along with their constrained solubility values. The task is regression on each graph to predict the constrained solubility values. For evaluation, we measure the mean absolute error on the predicted values.

**ALCHEMY:**

The Alchemy dataset Chen et al. (2019a) consists of nearly 200K graphs along with their 12 quantum mechanical properties. We chose the subset provided by Morris et al. (2020). The dataset comes with 12000 graphs. The test set and the validation set contain 1000 graphs each. The task is to regress on the 12 quantum mechanical properties and evaluate on mean absolute error.

**QM9**

QM9 Ramakrishnan et al. (2014) is a large scale molecular dataset consisting of nearly 130000 graphs. Each graph consists of a set of node features associated with atoms along with bonds encoded as edge features. Additionally, we are provided with 3D node position coordinates for each atom. The task is graph regression on a set of 12 quantum-chemical properties associated with each graph. We follow the convention of randomly splitting the dataset with 10000 graphs each in testing and validation set. The rest of the graphs are used for training. Interestingly, the node position coordinates have been shown to be very useful in Klicpera et al. (2020). However, we conduct two separate experiments, one with node position coordinates and other without the position features.

**OGBG-MOLHIV**

Ogbg-Molhiv dataset consists of 41127 molecular graphs. The task is binary classification for molecular property prediction. We use the full 9-dimensional node features, and adopt the standard scaffold split provided by Hu et al. (2020).

A.6 EXTENDED PRELIMINARIES

In this subsection, we provide an extended discussion on the graph coloring process and the search tree which PF-GNN is designed to approximate. For a detailed explanation, please refer to McKay & Piperno (2014). Below figures are adapted from McKay & Piperno (2021), a visual guide to the practical graph isomorphism solvers from the authors of Nauty and Traces.

**Coloring and 1-WL refinement:**

Graph Neural Networks have been shown to be neural analogues of vertex color refinement technique with neural embeddings being equivalent to the colors. Graph coloring is technique of assigning graph vertices with unique numbers in a permutation invariant manner; thereby ordering the vertices. 1-WL algorithm is one way coloring vertices and is also called naive-vertex refinement. In 1-WL

refinement, initially all nodes are assigned with the same color. In each iteration, 1-WL reassigns colors of each node based on its neighbourhood. As long as two vertices have different multiset of colors in their neighbourhood, they will be assigned different colors. However, after some iterations, the colors stop changing. At this point, the coloring is called *equitable coloring*.

### Equitable coloring

An equitable coloring is one where every two vertices of the same colour are adjacent to the same number of vertices of each colour. Fig. 4 shows some of the example equitable colorings. The colorings divides the set of vertices into disjoint sets based on the node colors. Hence, equitable colorings are also loosely called equitable partitions. An equitable coloring where each node has unique color is call *discrete coloring*.

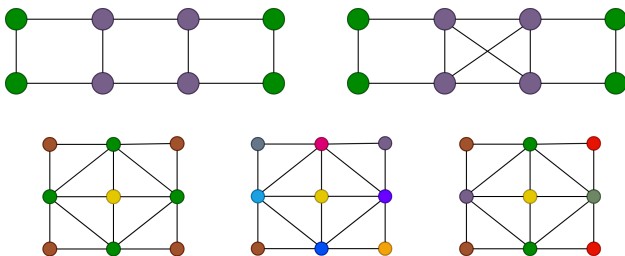

Figure 4: Example equitable partitions. The middle coloring in the second row is both equitable and discrete coloring.

### Identifying graphs:

In order to uniquely identify graphs, we need to color every vertex uniquely in a permutation invariant manner. For this, we need to generate discrete coloring in a permutation invariant way. Due to symmetry in the graph structures, equitable colorings do not allow 1-WL process to further refine the colorings. Therefore, GNNs are also limited in their expressive power to distinguish all non-isomorphic graphs. This bottleneck can be broken with a technique called *individualization and refinement*.

### Individualization and refinement:

Individualization and refinement is a technique of breaking the symmetry in the equitable colorings by recoloring a node with a unique color. Once, a node is distinguished from rest of the vertices, this information can be propagated the 1-WL/GNN message passing. Fig. 5 shows an example of the IR process. Initially a vertex is *individualized* by coloring it Green. Then in the refinement stage, its neighbours will be distinguished from other vertices. Thereafter, rest of the vertices are similarly distinguished iteratively till a finer equitable coloring is reached.

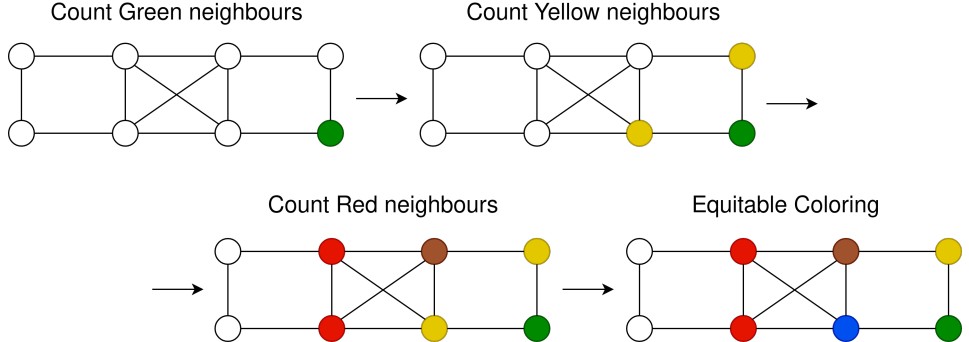

Figure 5: IR process: First a vertex is distinguished from rest of the vertices and this information is propagated to other nodes via message passing to generate a refined equitable coloring of the graph.

**Search tree**

The IR process is not a permutation invariant operation as the refined coloring depends on the individualized node in the graph. Therefore, in order to preserve permutation invariance, we need to repeat the IR process for all the nodes which have the same color *i.e.*, belonging to the same color cell in the equitable coloring. This gives rise to a search-tree where the equitable colorings form tree-nodes. The search tree is a unique representation of the isomorphism class of the input of graph *i.e.*, all isomorphic graphs generate the same search tree and non-isomorphic graphs generate different search trees. Fig. 2 shows the example of one such search tree.

