# OpenReview forum: "PF-GNN: Differentiable particle filtering based approximation of universal graph representations"
_ICLR.cc/2022/Conference — ICLR 2022 Poster_

### Official Review · Reviewer_YGAz · 2021-10-27

**Correctness:** 3
**Technical Novelty And Significance:** 3
**Empirical Novelty And Significance:** 3
**Recommendation:** 6
**Confidence:** 2

**Main Review:**

## Pros:

(+) The design and results are theoretically sound.

(+) The empirical results show that PF-GNN outperforms many existing GNN methods on various graph-level tasks.

## Cons:

(-) The clarity of the paper can be improved for non-expert readers.

## Detail comments:

I am not an expert in the graph isomorphism test and particle filter approach discussed in the paper. So, it is quite likely that I do not fully understand all the details. I get the high-level idea that the individualization approach improves the 1-WL test by breaking the symmetry. The experiment results demonstrate the superior performance of PF-GNN. Overall, I think it is a good paper, but the clarity and the way the authors explain their approach can be more friendly to non-expert readers.

One question that I have is that how PF-GNN performs compare to the standard graph isomorphism solver mentioned by the authors (such as Nauty and Traces?). Since PF-GNN is designed based on the IR approach, I would like to see such a comparison as a sanity check of whether we should use GNN-based approaches or just standard solvers. I understand that the embedding of GNN can be used for various downstream tasks and is not restricted to graph isomorphism tests. Yet such a comparison is desirable.

Finally, I notice that the authors demonstrate the time complexity for SR25 datasets compared to their backbone. However, I wonder what is the time complexity of PF-GNN on the other tasks and datasets such as ZINC, ALCHEMY, and QM9. How is the exact running time of PF-GNN to achieve those superior results in these datasets?


**Summary Of The Paper:**

The authors propose PF-GNN for graph-level tasks. Their design is based on exact isomorphism solver. They propose sampling process with particle filter updates to alleviate the high complexity issue.

**Summary Of The Review:**

The paper is theoretically sounded and the empirical performance of PF-GNN is impressive. Still, I think the explanation can be improved and I also have some regarding the experiment. Together I lean to vote for acceptance of this paper.

---

> ### Author Response · Authors · 2021-11-16
> **Author response to Reviewer YGAz**
>
> Thank you for your valuable comments and pertinent suggestions! Please see below for our responses to specific queries.
>
>
> + **Clarity and the way the authors explain their approach can be more friendly to non-expert readers.**
>
>     Thank you for the very pertinent suggestion on improving the readability of the paper to a wider audience. For the background on the isomorphism solvers and graph-coloring terminologies, we will include more explanations with pictures from https://pallini.di.uniroma1.it/Introduction.html. These illustrations are from the authors of Nauty/Traces[1]. Given the space constraints, these illustrations will be added in Appendix. For particle filters, we will further improve the clarity in our explanation.
>
>
> + **Since PF-GNN is designed based on the IR approach, a comparison of whether we should use GNN-based approaches or just standard solvers for graph isomorphism test.**
>
>     Whenever neural approaches are used to solve combinatorial problems like TSP, vertex cover etc., they are compared with the standard approximate solvers as well. Usually, the approximation ratios are used as a metric for comparison to see if a data-driven approach gives better approximate solutions. PF-GNN uses isomorphism solver techniques to learn better graph embeddings which can be used to approximately solve the isomorphism as well. However, note that isomorphism solvers like Nauty/Traces are exact solvers. They are 100% accurate on all graphs. Therefore, comparing PF-GNN with Nauty/Traces on graph isomorphism will not be informative as Nauty/Traces will always outperform on all graphs. We emphasize that the motivation to use isomorphism solving techniques in the design of PF-GNN is not to solve isomorphism but instead to enhance the expressive power of GNN which is often measured by its ability to distinguish non-isomorphiic graphs.
>
>
> + **Time complexity of PF-GNN on real-world datasets.**
>
>     Runtime of PF-GNN increases only linearly with the number of IR steps. Assuming GNN takes linear time for bounded-degree graphs O(n), PF-GNN with T steps of IR takes O(Tn). Note that the computation of K paths can be parallelized.
> Below we provide the empirical runtimes of PF-GNN models for different K and T values for the Zinc and Alchemy datasets. We also compare with the runtime of the standalone backbone GNN model. The numbers in brackets indicate the ratio of the runtime of PF-GNN w.r.t that of backbone GNN.
>     | Time (Ratio) | **GNN** | **K=4, T=1** | **K=4, T=2** | **K=4, T=3** | **K=8, T=1** | **K=8, T=2** | **K=8, T=3** |
>     |---|---|---|---|---|---|---|---|
>     | Alchemy | 6.72s (1) | 11.70s (1.74) | 14.82s (2.20) | 18.24s (2.71) | 12.28s (1.82) | 17.35s (2.58) | 23.57s (3.51) |
>     | Zinc | 8.42s (1) | 13.12s (1.73) | 19.24s (2.54) | 26.24s (3.48) | 17.14s (2.03) | 27.11s (3.21) | 38.57s (4.47) |
>
>
> References:
>
> [1] McKay, B.D. and Piperno  (2014), A., Practical Graph Isomorphism, II, Journal of Symbolic Computation, 60 pp. 94-112,

---

> > ### Comment · Reviewer_YGAz · 2021-11-18
> > **Re:**
> >
> > Thanks for the explanation. Most of my concerns are resolved. I understand that Nauty and Traces might always have 100% accuracy in the graph isomorphism test. However, as mentioned in the provided link, they are at best quasi-polynomial in time complexity. That is being said, maybe PF-GNN can be much more efficient in time compare to these exact solvers. As mentioned by the authors, many neural solutions to combinatorial problems are highlighted with their time efficiency (i.e. [1]). Even if PF-GNN is not as good as those exact solvers, it is still good to know the gap between using GNN approaches to these exact solvers. Nevertheless, I understand that this is not the main focus of this paper. Hence, I'm still positive to the work and keep my score unchanged.
> >
> >
> > ### References:
> >
> > [1] Exact Combinatorial Optimization with Graph Convolutional Neural Networks, Gasse et al. NeurIPS 2019.

---

> > > ### Author Response · Authors · 2021-11-21
> > > **Author Reply**
> > >
> > > Thank you very much for your positive response to our work! We are glad you found our responses satisfactory. We would look into the time efficiency of PF-GNN in comparison with graph isomorphism solvers. Moreover, in addition to isomorphism, using approximate universal embeddings to solve combinatorial problems on graphs seems an interesting direction for follow-up work on this.
> > >
> > > We have uploaded the revised version incorporating all the suggestions. Thanks again!

---

### Official Review · Reviewer_weJp · 2021-10-28

**Correctness:** 3
**Technical Novelty And Significance:** 2
**Empirical Novelty And Significance:** 2
**Recommendation:** 8
**Confidence:** 5

**Main Review:**

The paper introduces a novel, principled means to perform individualization, inspired by IR approaches within graph isomorphism solvers. Moreover, it conducts an extensive empirical analysis to evaluate its claims, and its reported results confirm the value of sequential sampling and highlight the strength of the model. Furthermore, this approach, much like random node initialization, can easily be applied across standard GNNs, which makes it relevant across the GNN community. Finally, the paper is clearly written and its main approach is well-presented. However, I have some concerns regarding the novelty and significance of the approach, which I enumerate below:

- The paper clearly explains and motivates its particle filtering mechanism, as well as its inspiration by IR. It also performs ablations on its individualization distributions and on its loss to highlight their significance. However, the paper does not consider ablations on its individualization method (via an MLP), or the means of inidividualization (partial or full randomization/individualization, as is studied in random node initialization works). This leaves a rather large gap between random node initialization and PF-GNN, from which any one (or many) distinguishing factors could be key contributors to the observed experimental differences. To illustrate, the difference between RNI and PF-GNN could mostly stem from sampling one node per iteration rather than all nodes initially, or it could stem from the learnable nature of the individualization in PF-GNN, as opposed to the fully randomized initialization in RNI (where the choice of initial distribution could also play a part). It could also be that partial RNI yields better performance as it preserves some deterministic information, as has been observed in Abboud et al (2020). Hence, it is not at all clear how or why PF-GNN achieves these improvements at present, and thus it is not sound to attribute these improvements solely to the PF mechanism and its data-driven approach based on current evidence.
- As mentioned earlier, the paper only compares with fully randomized RNI. This is not a fair comparison, in my opinion, as fully random RNI has no deterministic information to build on, whereas PF-GNN, by nature, preserves all but one node's features at any given iteration. Therefore, the paper should also consider partial RNI (at different percentages of randomization), where only a fraction of node embeddings is randomized, so as to better appraise the contributions of PF-GNN.
- The paper claims that PF-GNN enables better performance on real-world datasets, but does not use any standard benchmarks to validate this point. Hence, the authors should consider running experiments on standard benchmarks, such as those in OGB [1], to further corroborate this point.

[1] Hu et al., Open Graph Benchmark: Datasets for Machine Learning on Graphs. NeurIPS 2020.


**Summary Of The Paper:**

The paper introduces PF-GNN, a new method through which to individualize node representations with the aim of increasing GNN expressiveness. It is inspired by graph isomorphism algorithms' individualize and refine (IR) approaches, which individualize individual nodes then refine colorings, and then aggregate all colorings across the search tree of possible individualization paths. Specifically, PF-GNN approximately emulates IR by repeating a random sampling k times and returning an aggregate coloring from each sample. For every sample, T iterations are made, such that a sample node is selected for individualization, and then representations are updated using a GNN. However, PF-GNN introduces learnable components to this mechanism, such that affinities for node selection (yielding a distribution over all possible individualization candidate nodes) are learned, rather than simple uniform sampling. Furthermore, beliefs for sampling are iteratively updated in keeping with a particle filtering approach (PF), and node individualization itself is made using a learnable function, to enable more flexibility and data-driven individualization. Finally, the model is empirically evaluated on a series of datasets and compared with existing standard GNNs and individualization techniques, e.g., random node initialization (RNI). In these experiments, PF-GNN is shown to improve model expressiveness, is more resilient than RNI for larger numbers of nodes, and achieves strong performance on real-world datasets.

**Summary Of The Review:**

The paper proposes PF-GNN, an interesting addition to the literature on individualizing GNN node embeddings which demonstrates strong empirical performance. However, the value of the approach cannot be fully established, as the gap with existing methods is very large, implying that multiple factors (e.g., randomization techniques, initialization choices) could also be responsible for the performance improvement, rather than the proposed approach. Moreover, further evidence and experiments is required to validate the strength of PF-GNN over real-world data. Hence, the authors should conduct further experiments with intermediate randomization and individualization choices, and evaluate PF-GNN on standard benchmarks, to better quantify the impact of their proposed approach and confirm its significance.

DISCUSSION UPDATE: Rating increased from 5 to 8 following additional experiments (details in review replies).

---

> ### Author Response · Authors · 2021-11-16
> **Author response to Reviewer weJp (Part 1 of 2)**
>
> Thank you for your valuable and detailed comments! We have added suggested ablative experiments to further validate PF-GNN's components. Please see below for our responses to specific queries.
>
> + **Ablations on PF-GNN’s individualization method (via an MLP), or the means of individualization...**
>
>     We first study the importance of the individualization method on performance. In PF-GNN, individualization method involves transforming the selected node embedding via an MLP. In this ablation study, we replace the learnable MLP with non-learnable random normal features as used in RNI (Abboud et al. 2020) and compare the performances. We further compare both methods of individualization for various partial levels of node-feature perturbations as in RNI. For example, 50% refers to 50% embedding dimensions of the selected node are perturbed in each IR step. PF-GNN with T=3 and K=4 was used in the experiments.
>     | Dataset | MLP (100%) | MLP (87%) | MLP (50%) | MLP (12.5%) | random (100%) | random (87.5%) | random (50%) | random (12.5%) |
>     |---|---|---|---|---|---|---|---|---|
>     | Triangle-Large | 68.9 | 67.9 | 68.3 | 61.4 | 64.5 | 57.4 | 49.3 | 27.5 |
>     | CSL | 100 | 100 | 100 | 100 | 100 | 100 | 44.6 | 8.6 |
>
>     Above results suggest that learnable MLP is better than random perturbations for node individualization. For partial individualization, MLP performs similarly or better than random features. For full individualization, the difference is less clear in CSL dataset as both methods give similar result. However, very small levels of partial random perturbation harms the performance in both datasets.
>
>
> + **PF-GNN ablations on sequential individualization.**
>
>     We now study the impact of sequential individualization of PF-GNN. In the ablation study of the main paper, we show in Table 8 that sequential individualization without learning significantly reduces performance i.e. which nodes are selected in each IR step is important.
> We further examine the effects of sequential individualization compared to simultaneous individualization. For this, we compare following ablation models of PF-GNN: 1) PF-GNN with T=3 steps of IR, 2) PF-GNN-A: PF-GNN with a single IR step but 3 randomly chosen nodes individualized in the same step. 3) PF-GNN-B: single IR step but 6 random nodes are individualized at once 4) PF-GNN C: single IR step with 9 nodes individualized.
> Note that we individualize same number of nodes for PF-GNN method(sequential) and PF-GNN-A(3) method(simultaneous).
>     | Dataset |  PF-GNN  |   PF-GNN-A(3)  |   PF-GNN-B(6)    |   PF-GNN-C(9)   |
>     |---|---|---|---|---|
>     | Triangles-Large | 68.7 | 35.5 | 36.2 | 37.6|
>     | CSL | 100 | 44.67 | 36.67 | 26.67 |
>
>     - Above results on both the datasets show that sequentially individualizing selected nodes is better than individualizing a subset of nodes at once. Since most nodes are indistinguishable initially, it is hard to learn the optimal subset in one step. With sequential IR, the graph-coloring is refined in each step and the model can learn to successively pick nodes which are likely to generate the best refinement. Therefore, "Sequential IR process with learning" helps in finding those subset of nodes. Furthermore in CSL dataset, performance goes down as we increase the number of nodes individualized at once, suggesting that larger unguided randomness may not be helpful. Overall, the results suggest that PF-GNN's data-driven sequential refinement improves GNN's expressivity while adding minimal randomness.
>
>
> + **Comparison of PF-GNN with Random Node Initialization (RNI) and partial RNI.**
>
>     We report both full and partial RNI results on the CSL, LCC and TRIANGLES datasets. The results show that PF-GNN outperforms both full and partial RNI. Although RNI improves over GIN, partial RNI does not seem to help in the reported datasets. A possible reason for these results could be that partial RNI helps when the dataset contains informative node features. However, the synthetic datasets we have used did not contain any node features and nodes were instantiated with constant values. Hence, there is no deterministic information which can be exploited by GNN with partial RNI. It would be interesting to further explore the conditions under which partial RNI would perform better than RNI.
>
>     Additionally, RNI methods are slow to converge (Abboud et al 2020) and need more data to generalize while PF-GNN uses minimal randomness and can generalize better than RNI.
>
>     | Dataset | PF-GNN | GIN | RNI(100%) | RNI(87.5%) | RNI(50%) | RNI(12.5%) |
>     |--|---|--|--|--|--|--|
>     | Triangles-O | 99 | 47 | 59 | 54 | 52 | 49 |
>     | Triangles-L | 72 | 18 | 31 | 29 | 27 | 27 |
>     | LCC-N | 100 | 50 | 82 | 82 | 83 | 82 |
>     | LCC-X | 100 | 50 | 87 | 88 | 88 | 87 |
>     | CSL | 100 | 10 | 32.67 | 34 | 35.33 | 34.67 |
>
> References:
>
> Abboud et al (2020) "The surprising power of graph neural networks with random node initialization" arXiv:2010.01179

---

> > ### Author Response · Authors · 2021-11-16
> > **Author response to Reviewer weJp (Part 2 of 2)**
> >
> > + **Performance of PF-GNN on OGB datasets**
> >
> > We report results of PF-GNN on OGB-molhiv dataset below. We will share the hyperparameter details in the updated paper. The results suggest that PF-GNN shows competitive performance on the OGB-molhiv data set. These results are without using fingerprint features.
> >
> > | Method | GIN | DeeperGCN | PNA [2] | DGN [3] | Direction GSN [4] | PF-GNN |
> > |---|---|---|---|---|---|---|
> > | ROC-AUC | 75.58 | 78.58 | 79.05 | 79.7 | 80.39 | 80.15 |
> >
> >
> > References:
> >
> > [1] Hu et al., Open Graph Benchmark: Datasets for Machine Learning on Graphs. NeurIPS 2020.
> >
> > [2] Corso et al. "Principal Neighbourhood Aggregation for Graph Nets." Neurips (2020).
> >
> > [3] Beani  et al. "Directional graph networks." ICML, 2021.
> >
> > [4] Bouritsas et al. "Improving graph neural network expressivity via subgraph isomorphism counting." arXiv:2006.09252 (2020).

---

> > > ### Comment · Reviewer_weJp · 2021-11-17
> > > **Reviewer Response**
> > >
> > > Thanks very much for the detailed rebuttal, and for running the additional experiments. The picture looks clearer now. I do, however, have some follow-up comments:
> > > - Could you elaborate more on how the randomization experiments are performed? In particular, what initialization distribution(s) are used, and have you noticed any distributional differences (if applicable)? Also, are the reported numbers following convergence of the RNI models, or was there a fixed training cutoff point? In other words, how were these models trained? I revisited the paper to check the specifics of your training setup, and only saw one mention of the number of training epochs in the caption of Table 5.1 (100 epochs). Could you please confirm whether models were trained until convergence, or whether a fixed cutoff was used as in Section 5.1? If the latter, then I suggest rerunning these experiments until convergence. I understand that this will be slower to converge, but only such a study can fully answer the question of whether the MLP (or the approach in general) improves performance irrespective of training duration.
> > > - It does make sense that setting vectors in a principled, sequential manner will yield improvements, and I am satisfied with how this experiment is conducted. The same question about training loop also applies here, though.
> > > - Real-world experimental results are also as expected. The model remains competitive, and this is reassuring, as randomization runs the risk of eliminating useful inductive biases. This seems to have been avoided here. It is also understandable for randomization not to improve performance in my opinion, as real-world data instances are highly unlikely to exhibit challenging (from an expressiveness perspective) instances, making the expressiveness gain unlikely to mainfest into better results.
> > >
> > > All in all, I am happy with the response, and am willing to increase my rating once the above concerns have been addressed.

---

> > > > ### Author Response · Authors · 2021-11-19
> > > > **Author response to follow-up questions of Reviewer weJp**
> > > >
> > > > Thank you very much for the response. We are glad you found the additional experiments satisfactory! Please find below our response to the follow-up queries.
> > > >
> > > > + **Regarding random initialization distributions**
> > > >
> > > >     We initially performed experiments with random uniform and normal distributions. However, we did not find considerable difference in performance between the two distributions. Therefore, we conducted all the reported  experiments on random normal features as it was shown to be slightly better in Abboud et al (2020).
> > > >
> > > >
> > > > + **Training till fixed cutoff point or till convergence**
> > > >
> > > >     For the above reported results on randomization, we trained all models for 500 epochs for all three datasets.
> > > > For the TRIANGLES dataset, we observed that in almost all experiments, there is hardly any improvement after 150-200 epochs. Probable reason for this convergence in relatively less number of epochs would be that the TRIANGLES dataset is a large dataset with 30000 graphs in the training set while the other synthetic datasets are relatively very small. Therefore, it would be fair to say these results are well converged results. For CSL and LCC datasets, we repeated all the experiments till convergence and report the updated results below. Overall, on average the accuracies are slightly increased after running till convergence. However, the results exhibit similar patterns noted previously for partial/full individualization as well as sequential individualization and RNI.
> > > >
> > > >     - **Ablations on PF-GNN’s individualization method (via an MLP), or the means of individualization…**
> > > >
> > > >         | Dataset | MLP(100%) | MLP(87%) | MLP(50%) | MLP(12.5%) | random (100%) | random (87.5%) | random (50%) | random (12.5%) |
> > > >         |---|---|---|---|---|---|---|---|---|
> > > >         | Triangle-Large | 68.9 | 67.9 | 68.3 | 61.4 | 64.5 | 57.4 | 49.3 | 27.5 |
> > > >         | CSL | 100 | 100 | 100 | 100 | 100 | 100 | 82.6 | 48.0 |
> > > >
> > > >     - **PF-GNN ablations on sequential individualization.**
> > > >
> > > >         | Dataset | PF-GNN | PF-GNN-A(3) | PF-GNN-B(6)     | PF-GNN-C(9) |
> > > >         |---|---|---|---|---|
> > > >         | Triangles-large | 68.7 | 35.5 | 36.2 | 37.6|
> > > >         | CSL | 100 | 49.23 | 42.67 | 41.33 |
> > > >
> > > >     - **Comparison of PF-GNN with Random Node Initialization (RNI) and partial RNI.**
> > > >
> > > >         | Dataset | PF-GNN | GIN | RNI(100%) | RNI(87.5%) | RNI(50%) | RNI(12.5%) |
> > > >         |---|---|---|---|---|---|---|
> > > >         | Triangles-O | 99 | 47 | 59 | 54 | 52 | 49 |
> > > >         | Triangles-L | 72 | 18 | 31 | 29 | 27 | 27 |
> > > >         | LCC-N | 100 | 50 | 82 | 82 | 83 | 83 |
> > > >         | LCC-X | 100 | 50 | 89 | 90 | 86 | 86 |
> > > >         | CSL | 100 | 10 | 38.67 | 38.67 | 38.67 | 36.00 |
> > > >
> > > >
> > > > + **Real-world experimental results**
> > > >
> > > >     We agree that for real-world datasets, it is hard to improve upon state-of-the-art results as these datasets probably do not have much structure which need to be captured by more expressive models. Instead, models  with better inductive bias towards the end-task seem to perform better. However, we do note that PF-GNN improves upon the base 1-WL GNNs considerably on molecular datasets like Zinc and Alchemy. A probable reason seems that molecules may have more symmetry issues as compared to other types of real-world graph datasets, since the atoms tend to come from a small number of types and graphs usually contain repeating patterns of substructures. It would be interesting to further try to understand the kind of symmetry issues molecules exhibit.

---

> > > > > ### Comment · Reviewer_weJp · 2021-11-21
> > > > > **Reviewer Response**
> > > > >
> > > > > Thanks very much for your clarifications. I will increase my rating.

---

> > > > > > ### Author Response · Authors · 2021-11-21
> > > > > > **Author reply**
> > > > > >
> > > > > > Thank you very much for increasing the score!! We appreciate your suggested experiments which helped us improve the paper.

---

### Official Review · Reviewer_tTwv · 2021-11-01

**Correctness:** 3
**Technical Novelty And Significance:** 4
**Empirical Novelty And Significance:** 3
**Recommendation:** 6
**Confidence:** 5

**Main Review:**

Strengths:
1. Adapting IR to neural and GNN area is a novel and important contribution. The designed algorithm does improve over base GNNs.
2. The particle filtering algorithm is a elegant and low-complexity realization of IR algorithm.

Weakness:
1. Particle filtering samples K paths to approximate summation of exponential number of paths in original IR which is theoretically universal in graph isomorphism test, it's would be great to show how much expressive do the K paths keep, with changing K from 1 to K_max (like 30). Nevertheless it's hard to believe K paths are enough to cover exponential number of paths.
2. Particle filtering assumes all intermediate observations are given but the designed algorithm doesn't have any intermediate observations. The author may need to provide further justification for this significant misalignment.
3. A bad property for sampling method is that for two same graphs, the algorithm may judge them as non-isomorphic. This can hurt generalization ability a lot. It would be great to figure out ways to solve the issue.
4. Although not a main issue, the theorem 2 proof seems  incorrect as the value is independent of T. Eq (22) -> Eq (23) is not clear.
5. The major weakness is the scalability and practical complexity. The designed model is T times deeper and larger than base model, and K path sampling needs K times larger memory for parallel computation. For ZINC dataset I have run the author's implementation and it's too slow comparing with base GNN also eats too much memory. Making the proposed algorithm more practical is extremely important for larger dataset like CIFAR10.

**Summary Of The Paper:**

This paper presents a neural version of individual-refinement (IR) architecture for improving the expressiveness of GNN in terms of isomorphism test. As IR is the dominant approach of practical graph isomorphism test, adapting IR to GNN is a novel and important idea. As IR suffers from the exponential number of branches, the paper adapts particle filtering algorithm to sample K paths to approximate the full version of IR algorithm.  Simulation and real-world datasets are used to demonstrate the improvement over base GNN.

**Summary Of The Review:**

Overall I think the paper deserves a weak accept: the proposed idea of generalizing IR is novel and important, however the method is still not practical enough to use, and without good theoretical theorem for the expressiveness of the designed sampling version IR.

---

> ### Author Response · Authors · 2021-11-16
> **Author response to Reviewer tTwv (Part 1 of 2)**
>
> Thank you for your valuable feedback and insightful comments! Please see below for our responses to specific queries.
>
> 1. **..it's hard to believe K paths are enough to cover exponential number of paths.**
>
>     The number of paths (K) needed to represent the search-tree can be small whenever there is large redundancy in the search-tree i.e. colorings are repeated at each level of the tree. This makes the tree have small number of unique paths from root to the leaves. The redundancy is due to the presence of automorphisms in the graph. Usually, graphs which have high symmetry in their structure contain vertices that can be mapped to each other via some automorphism. These vertices are structurally equivalent, hence induce exactly the same refinement upon individualization. For example, in Figure 2, there is a single unique coloring at level-3 and all the leaves are rotated versions of the same coloring. Exact isomorphism solvers like Nauty/Traces find the automorphisms on the fly and prune the search-tree. The pruning of subtrees makes the solvers practically very fast; even though they can take exponential time in the worst case. Therefore, on account of the large redundancy in the search-trees, a small number of paths are likely to be sufficient to get good embeddings with less variance. Our experimental results support this as increasing K after 4 or 8, does not help much in improving the performance significantly in most of our reported datasets. Additionally, reweighting with observation function combined with resampling has similar effect to pruning away subtrees in the search-tree, further reducing the required K.
>
>
> 2. **Particle filtering assumes all intermediate observations are given but the designed algorithm doesn't have any intermediate observations.**
> In particle filtering algorithm, a generative model $p(o|x)$ is used to up-weight or down-weight the particles in each step. Following Ma et al. (2020), we have used a discriminative model $f_{obs}(x;\theta)$ instead, which is trained to optimize the end task. $f_{obs}(x;\theta)$ is trained to measure the compatibility of the state with the prediction target and hence, we can up-weight/down-weight the state based on $f_{obs}(x;\theta)$. In essence, the discriminatively trained $f_{obs}(x;\theta)$ implicitly performs the same function as the generative model $p(o|x)$ in particle filters in the absence of the intermediate observations.
>
>
> 3. **sampling method may judge isomorphic graphs as non-isomorphic. This can hurt generalization ability a lot.**
> In any sampling based method, the best way to improve generalization in downstream tasks is to reduce variance in the embeddings. With less variance, even if the non-isomorphic graphs are very close, their embeddings will be less likely to cross into each other's space. Our approach reduces variance in multiple ways:
>     - Our individualization function (that analogously selects the color in the graph isomorphism solver) is learnt end-to-end. Learning will adapt the individualization function that reduces variance if it helps the end performance.
> 	- As discussed in previous comment, automorphisms result in many leaves being the same in the search-tree. This reduces the variance in the average embedding.
> 	- In each step, reweighting embeddings with the learnt observation function followed by resampling has similar effect to pruning away subtrees in the search-tree. By reducing the number of non-pruned leaves in the tree, the variance can again be reduced.
>     - Note that larger K and smaller T helps in reducing the variance in graph embeddings. Hence, we can further reduce variance by tuning the hyperparameters T and K to fit the dataset for optimal performance.
>
>         Finally, there is much scope for interesting future work on designing better loss functions to restrict the variance of generated embeddings.
>
>
> 4. **Theorem 2 proof seems incorrect as the value is independent of T. Eq (22) -> Eq (23) is not clear.**
>
>     In Theorem 2, we only consider sampling independently from the leaves of the search-tree. We can follow the “Ancestral/forward sampling” approach  in order to get independent samples from the distribution over the leaves of the tree. This does not involve T in the bound. However, if we consider resampling at every step, then the bound would involve T. This would need further assumptions on the behaviour of the  individualization and refinement functions. In follow up work, we plan to further study the analysis of the algorithm.
>
>     For Eq (22) -> Eq (23): We first consider the error between a single scalar value and its mean. Next we consider D-dimensional embedding and bound the error for all D values simultaneously with max-norm. For this, we add the error probability D times by using the “union bound” of probability. We will add one more step before Eq (23) for clarity.
>
> References:
>
> [1] Ma et al. (2020) Particle filter recurrent neural networks. AAAI 2020 arXiv:1905.12885

---

> > ### Author Response · Authors · 2021-11-16
> > **Author response to Reviewer tTwv (Part 2 of 2)**
> >
> > 5. **The major weakness is the scalability and practical complexity...**
> >
> >     While we agree that PF-GNN is slower in comparison to base GNN, the increase in runtime is only linear in the number of IR steps T. Assuming GNN takes linear time $O(n)$ for bounded-degree graphs, PF-GNN with T steps of IR takes $O(Tn)$ time with parallel computation of K paths. Also, note that a small T seems sufficient for real-world datasets as seen in our experiments. We believe this is a positive point for a more expressive GNN model since other expressive models like K-WL GNNs have runtime of $O(n^K)$.
> >
> >     The submitted code is not optimized for multiple GPUs, hence seems slower with more memory. For a fair comparison, we report runtime results on Zinc/Alchemy and compare it to the base GNN with same batch size on a single gpu.
> > Running the submitted code with batch size of 64 without the “--parallel” flag would reproduce results below. Runtime is in seconds per epoch and number in parentheses is the ratio compared with base GNN.
> >     | Time (Ratio) | **GNN** | **K=4, T=1** | **K=4, T=2** | **K=4, T=3** | **K=8, T=1** | **K=8, T=2** | **K=8, T=3** |
> >     |---|---|---|---|---|---|---|---|
> >     | Alchemy | 6.72s (1) | 11.70s (1.74) | 14.82s (2.20) | 18.24s (2.71) | 12.28s (1.82) | 17.35s (2.58) | 23.57s (3.51) |
> >     | Zinc | 8.42s (1) | 13.12s (1.73) | 19.24s (2.54) | 26.24s (3.48) | 17.14s (2.03) | 27.11s (3.21) | 38.57s (4.47) |
> >
> >     Furthermore, PF-GNN is a flexible model. We can reduce the number of gnn iterations within each IR step, if we want to keep the same time as the base GNN. Even so, we can still get performance improvements with PF-GNN. Below we report results of the standalone base GNN model with 6 iterations and PF-GNN with T=1 and T=2 but 6 gnn steps in total. Below results show that PF-GNN improves upon GNN with the same budget of total GNN layers.
> >
> >     |  | Zinc (mae) | Zinc (time) | Alchemy (mae) | Alchemy (time) |
> >     |---|---|---|---|---|
> >     | 6-step base GNN | 18.14 | 8.92s | 15.75 | 6.82s |
> >     | PF-GNN T=1 (gnn steps [4,2]) | 13.37 | 12.33s | 11.66 | 9.06s |
> >     | PF-GNN T=2 (gnn steps [2, 2, 2]) | 14.01 | 17.57s | 11.73 | 11.17s |

---

> > > ### Comment · Reviewer_tTwv · 2021-11-24
> > > **Response**
> > >
> > > Thank you for some additional clarification and experimental result, however the author hasn't solved my main issues.
> > >
> > > 1. Lacking the theoretical analysis for the expressiveness of k-path method.
> > > 2. For my third concern "sampling method may judge isomorphic graphs as non-isomorphic", I wish the author can provide some experimental result to show the effectiveness of different approaches mentioned by the author.
> > > 3. For my last concern, I wish the author can provide result over CIFAR10 or other larger graphs (you have to show that you can successfully train over these datasets with a reasonable runtime and memory cost), as molecular graphs are very small.

---

> > > > ### Author Response · Authors · 2021-11-26
> > > > **Author response to follow-up queries of Reviewer tTwv (Part 1 of 2)**
> > > >
> > > > Thank you very much for the response! We are glad some of your concerns are addressed. Below we address the follow-up queries.
> > > >
> > > >  1. **"..show how much expressive do the K paths keep, with changing K from 1 to K_max (like 30)”**
> > > >
> > > >     The K paths do not increase the expressivity of the model. Expressivity is increased by IR steps which refine the colorings. In fact, it is possible to get fully expressive embeddings even with a single path. As shown in Mckay et al (2014), there always exists a single path which gives a “canonical coloring” for all graphs. However, finding that path is expensive and we use sampling to efficiently approximate the set of all paths.
> > > >
> > > >     For long enough T to generate discrete coloring, Theorem 2 provides theoretical analysis on the number of paths required to get embeddings sufficiently close to mean when resampling is not done. Tighter bounds may be possible but require more assumptions, e.g. we can use Bernstein's inequality instead of Hoeffding's inequality but it would require assumptions on the variance of the leaf embeddings.
> > > >
> > > >     Bernstein's inequality: $P(\sum_{i=1}^{n} X_i > t) \leq \exp (-\frac{\frac{1}{2}t^2}{\sum_{i=1}^n \mathbb{E}[X_i^2] + \frac{1}{2}Mt})$
> > > >
> > > >     References:
> > > >
> > > >     McKay et al. (2014), A., Practical Graph Isomorphism, II, Journal of Symbolic Computation, 60 pp. 94-112,
> > > >
> > > >
> > > > 2. **Experiments for approaches to reduce variance - “sampling method may judge isomorphic graphs as non-isomorphic”**
> > > >
> > > >     We have provided experiments in the revised version of the paper which support three of the points mentioned in our response for reducing the variance. Note that lesser variance in embeddings can be reflected by gains in performance for the ablation experiments. Specifically,
> > > >
> > > >    1. For “an end-to-end learnable individualization function” :
> > > >
> > > >         In Section A.5.1, we empirically compare learnable individualization function with non-learnable random individualization. In principle, both MLP/random perturbation would achieve the individualization of the node.
> > > >     However, the results in Table 10 suggest that learning the individualization function has considerable benefit on the performance compared to non-learnable function. The benefit of using MLP appears to suggest that learbable individualization function helps in getting embeddings which generalize better.
> > > >
> > > >     2. For “reweighting with resampling similarity to pruning in IR solvers”:
> > > >
> > > >         In Table 9 of Section 5.4, we compared the ablation model of PF-GNN without the resampling step. Although the difference in performance is not large, it does show that resampling helps in improving accuracy.
> > > >
> > > >     3. For “larger K and smaller T helps in reducing the variance”:
> > > >
> > > >         In Table 8 of Section 5.4, we provide ablation results on the effects of increasing K and T on performance. Specifically, we can observe that for a fixed T, the performance increases as we increase K before saturating. However, for a fixed K, increasing T after 3 IR steps decreases performance.
> > > >
> > > >         This can be explained as higher values of T increases randomness and higher values of K reduces variance. Therefore, proper tuning of the hyperparameters K and T with a validation set can help in achieving a balance between expressivity(T) and less variance(K).
> > > >
> > > >     4. As for the presence of automorphisms, it requires further analysis on the number of automorphisms that are present in the graph. However, finding all the automorphisms is at least as hard as the problem of graph isomorphism itself. Therefore, we leave it for future work.

---

> > > > > ### Author Response · Authors · 2021-11-26
> > > > > **Author response to follow-up queries of Reviewer tTwv (Part 2 of 2)**
> > > > >
> > > > > 3. **Runtime and memory**:
> > > > >     In theory, the runtime of PF-GNN  is $O(nT)$ and the space requirement is $O(nK)$ for parallel computation of K paths. Hence, we emphasize that increase in time and memory is only linear in T and K.
> > > > >
> > > > >     **On CIFAR10 dataset**: We run the experiments for the CIFAR10 dataset as suggested. We compare training time per epoch of PF-GNN w.r.t. base GNN for different values of K and T. We run all experiments on a single GeForce RTX 2080 Ti GPU of 12GB memory. The batch size used is 64 and the base GNN used is GIN.
> > > > >
> > > > >     | Cifar-10        | GIN     | K=4, T=1 | K=4, T=2 | K=4, T=3 | K=8, T=1 | K=8, T=2 | K=8, T=3  |
> > > > >     |-----------------|---------|----------|----------|----------|----------|----------|-----------|
> > > > >     | Time            | 12.97s  | 26.12s   | 34.24s   | 46.97s   | 29.84s   | 49.28s   | 60.68s    |
> > > > >     | Ratio w.r.t GIN | 1       | 2.01     | 2.63     | 3.62     | 2.3      | 3.79     | 4.67      |
> > > > >
> > > > >     The results suggest a linear increase in runtime as we increase T, while the increase in runtime is marginal with the increase in K because of parallelization.
> > > > >
> > > > > 	**Memory**:
> > > > > The increase in memory requirement of PF-GNN is primarily due to parallel computation of K paths. Hence, we tested if we get Out of Memory (OOM) error for different values of K.
> > > > >
> > > > >
> > > > >     We are able to run all the above experiments without any OOM error. In fact, we are able to run without OOM for the model upto K=32 paths with T=3 steps and batch size of 64,
> > > > >
> > > > >     Overall, the above results suggest that we can train a larger dataset like CIFAR10 for various values of K and T within reasonable time and memory requirements.

---

> > > > > > ### Comment · Reviewer_tTwv · 2021-11-28
> > > > > > **Response to author**
> > > > > >
> > > > > > I appreciate the author's further explanation. I still have some concerns regarding the same issues. I will keep my score for now, mainly because the method and theory still have some issues.
> > > > > >
> > > > > > 1. Theorem 2 doesn't give a good justification of the expressivity of K-path method. 1) The bound is too loose: with the big O notation, the right part result doesn't give me any useful information for how to chose K. The only thing I get is that the K should be large enough which is obvious. 2) The main problem is that K-path method is definitely not universal, and theoretically characterize its expressiveness needs further investigation for a fixed K: for example fix K as 10 (what is the expressiveness of the 10-path method in terms of isomorphism test, how to describe it precisely if not connect with WL test?).
> > > > > >
> > > > > > 2. For the experiments for addressing “sampling method may judge isomorphic graphs as non-isomorphic”, I wish the author can directly reply to the question by designing some experiments to test that how many times the method will judge two same graphs as non-isomorphic. For example, the author can simulate N non-isomorphic graphs, train N-class classification with the method. After training, for each graph, do evaluation multiple times (input the same graph to the model multiple times as there is randomness) for the graph and report the percentage of times having same result.
> > > > > >
> > > > > > 3. The experiments on CIFAR10 is very helpful. However we know that there is a tradeoff between performance and resource: with decreasing K and T the performance will be worse. So it's essential for the author to report the performance comparison to see whether the method works with small K and T. Also, please always report the true memory cost for all methods: this is essential.

---

> > > > > > > ### Author Response · Authors · 2021-11-30
> > > > > > > **Author response to follow-up queries of Reviewer tTwv**
> > > > > > >
> > > > > > > Thank you very much for your further engagement. On account of limited time, we are not able to run all suggested experiments. Please find below our limited response to the follow-up queries.
> > > > > > >
> > > > > > > 1. **Analysis**
> > > > > > >
> > > > > > >     Thanks for the suggestion on additional analysis. We will work on further tighter analysis of the method and hopefully, update the revised version with a tighter bound on K.
> > > > > > >
> > > > > > > 2. **Experiments for - "sampling method may judge isomorphic graphs as non-isomorphic”**
> > > > > > >
> > > > > > >     As suggested, we ran experiments on CSL dataset to test the consistency of embeddings in test time. CSL dataset consists of 150 graphs belonging to 10 isomorphism classes. We train for 300 epochs and report the accuracy on the test set for 20 runs after each training. The below results are averaged over 5 runs.
> > > > > > >
> > > > > > >     | mean  | median |  max |  min     |    std    |
> > > > > > >     |----------|------------|--------|-----------|-----------|
> > > > > > >     | 98.84  | 100       | 100   |  90.00 |  2.47    |
> > > > > > >
> > > > > > >
> > > > > > >     For the CSL datasets,  we find that almost all the time, the classification accuracy stays near optimal with a mean of 98.84. However, the error rate could be different for different datasets. We will add further results on multiple datasets.
> > > > > > >
> > > > > > > 3. **Experiments on CIFAR10**
> > > > > > >
> > > > > > >     We reported the training time on CIFAR10 without optimising for performance. Unfortunately, due to limited time, we are not able to reproduce reported performance on many base GNNs on top of which PF-GNN could be applied. We will update once we are able to reproduce the results.  Below we provide the exact memory taken by various models. All experiments run with batch size = 64 and hidden dimension = 80 on a GPU of memory 12GB.
> > > > > > >
> > > > > > >     | Cifar-10        | GIN  | K=4, T=1 | K=4, T=2 | K=4, T=3 | K=8, T=1 | K=8, T=2 | K=8, T=3  |
> > > > > > >     |-----------------|------|----------|----------|----------|----------|----------|-----------|
> > > > > > >     | Memory (MB)     | 1165 | 1675    | 2129     | 2577     | 2376     | 3141     | 3865      |
> > > > > > >     | Ratio w.r.t GIN | 1    | 1.43     | 1.82     | 2.21     | 2.03     | 2.69     | 3.31      |
> > > > > > >
> > > > > > >     As previously noted, the memory requirement is not large for various values of K and T. The small increase in memory as we increase T includes additional layers of GNN. The increase in memory as we increase K is because of K copies of graph embeddings and does not lead to increase in parameters.

---

> > > > > > > > ### Comment · Reviewer_tTwv · 2021-11-30
> > > > > > > > **Final response**
> > > > > > > >
> > > > > > > > As a reviewer I appreciate the author's additional work on partially addressing these problems. I encourage the author to  continue improve the paper (or in another paper) on this direction, it would be really helpful for the community. My current thought is between accept and weak accept, mainly because the author's hard work. However I will keep weak accept as the paper needs another voice for future improvement.

---

> > > > > > > > > ### Author Response · Authors · 2021-11-30
> > > > > > > > > **Author reply**
> > > > > > > > >
> > > > > > > > > Thank you very much for your positive opinion on our work! We appreciate your thoughtful comments and pertinent suggestions during the discussion. Best regards.

---

### Official Review · Reviewer_WUrz · 2021-11-02

**Correctness:** 4
**Technical Novelty And Significance:** 3
**Empirical Novelty And Significance:** 3
**Recommendation:** 8
**Confidence:** 3

**Main Review:**

Summary:\
The paper is well written and easy to follow. In my opinion, the originality of the paper is high since it is both technically rich and the empirical results are strong. Although there are other papers that have proposed models that are more powerful than 1-WL, the approach proposed in this paper is different from prior work. The model is found to be very effective in graph isomorphism testing and in the detection of graph properties, while it outperforms  the baselines on three real-world datasets. Overall, the paper seems to be proposing a novel contribution.

Theoretical Motivation:\
The paper seems to have a solid theoretical motivation. First, the authors formulate the universality of the graph representations under the framework of individualization and refinement. Then, they show how the distance of the generated embeddings is approximated by sampling a number of paths from the search tree. I am not sure though how useful the bound provided in Theorem 2 could be in practice. In most experiments, the authors have set K equal to 4 or 8. For reasonable values of M and D, a much larger K would be required such that the embeddings of two graphs are separated with high probability by some distance close to the exact distance d.

Since the model performs sampling, to my understanding, two isomorphic graphs will obtain different embeddings from each other. These embeddings might be very similar to each other, however, they will not be identical to each other. Could this be a problem for some application (for instance if nearly isomorphic graphs belong to different classes or their targets in some regression problem are very different from each other)?

Experimental evaluation:\
The authors perform an extensive empirical evaluation of PF-GNN on both synthetic and real-world tasks. In both cases, their model exhibits a strong performance on learning expressive graph representations.
- One of my concerns with this paper is that PF-GNN is mainly evaluated on synthetic datasets, while only three real-world datasets are employed in total. It is not thus entirely clear how effective the proposed model is in real-world scenarios. I would suggest the authors evaluate also the model on standard graph classification datasets such as ENZYMES, NCI1, IMDB-BINARY, REDDIT-BINARY, etc or on some graph property prediction dataset from OGB.
- With regards to the synthetic datasets, the authors could also investigate whether the proposed model can identify fundamental graph properties such as bipartiteness, connectivity and planarity (this problem has been studied in [1] in the context of graph kernels).

Minor comments:\
Please explain how the PF-GNN model is trained on the three graph isomorphism test datasets. What are the class labels of the different samples?

In Table 7, it is shown that a larger number of IR steps does not necessarily lead to a higher classification accuracy. What is the reason behind that? I would like the authors to provide some explanation.

Typos:\
"consistently outperform" => "consistently outperforms"

References:\
[1] A Property Testing Framework for the Theoretical Expressivity of Graph Kernels. Nils M. Kriege et al, In IJCAI'18.


**Summary Of The Paper:**

This paper proposes an extension of message passing GNNs that can learn approximately universal graph representations. Specifically, based on the colourings search tree method of exact isomorphism solvers, the authors suggest a differentiable approximation over the search tree by sampling multiple paths.

**Summary Of The Review:**

The paper proposes an original contribution for the graph representation learning community. The proposed model is novel, while the paper provides some theoretical analysis of its benefits. The empirical results are strong since the PF-GNN model outperforms the baselines on almost all datasets.

---

> ### Author Response · Authors · 2021-11-16
> **Author response to Reviewer WUrz**
>
>
> Thank you for your valuable feedback and insightful comments! Please see below for our responses to specific queries.
> + **Regarding Theorem 2 and the value of K needed to separate non-isomorphic graphs.**
>
>     The bound in Theorem 2 is not tight, since it does not take into account the resampling at each level as well as the presence of automorphisms in the graph. Usually, graphs which have high symmetry in their structure contain vertices that can be mapped to each other via some automorphisms. These vertices are structurally equivalent, hence induce exactly the same refinements upon individualization. This leads to a large number of repeated colorings at different branches of the search-tree. For example, in Figure 2, there is a single unique coloring at level-3 and all the leaves are rotated versions of the same coloring. Exact isomorphism solvers like Nauty/Traces find the automorphisms on the fly and prune the search-tree. The pruning of subtrees makes the solvers practically very fast; even though they can take exponential time in the worst case. Therefore, due to large redundancy in the search-trees, a small number of paths are likely to be sufficient to get good embeddings with less variance. Our experimental results support this as increasing K after 4 or 8, does not help improve the performance significantly in most of our reported datasets. Additionally, reweighting with observation function combined with resampling has similar effect to pruning away subtrees in the search-tree, further reducing the required K.
>
>
> + **Since the model performs sampling, isomorphic graphs will get different embeddings which may hurt prediction in downstream applications..**
>
>     In any sampling based method, the best way to improve generalization in downstream tasks is to reduce variance in the embeddings. With less variance, even if the non-isomorphic graphs are very close, their embeddings will be less likely to cross into each other's space. Our approach reduces variance in multiple ways:
>     - Our individualization function (that analogously selects the color in the graph isomorphism solver) is learnt end-to-end. Learning will adapt the individualization function that reduces variance if it helps the end performance.
> 	- As discussed in previous comment, automorphisms result in many leaves being the same in the search-tree. This reduces the variance in the average embedding.
> 	- In each step, reweighting embeddings with learnt observation function followed by resampling has similar effect to pruning away subtrees in the search-tree which again reduces variance.
>     - Finally, larger K and smaller T helps in reducing the variance. Hence, we can further reduce variance by tuning the hyperparameters T and K to fit the dataset for optimal performance.
>
> + **Experiments on more real datasets and graph properties such as bipartiteness, connectivity and planarity**
> We report results of PF-GNN on OGB-molhiv dataset and on synthetic datasets for the task of predicting three graph properties: connectivity, bipartiteness and planarity (Kriege et al. 2018). We will share experimental details in the updated paper. The results suggest that PF-GNN shows competitive performance on OGB-molhiv and achieves near optimal accuracy on the graph properties well above GIN. Note that all OGB-molhiv results are without using fingerprint features.
>     - OGB-molhiv :
>       | Method | GIN | DeeperGCN | PNA | DGN | Direction-GSN | PF-GNN |
>       |---|---|---|---|---|---|---|
>       | ROC-AUC | 75.58 | 78.58 | 79.05 | 79.7 | 80.39 | 80.15 |
>     - Graph-properties :
>       | Method | **Connectivity** | **Planarity** | **Bipartiteness** |
>       |---|---|---|---|
>       | GIN | 62.4 | 50.5 | 55.7 |
>       | PF-GNN | 97.5 | 98.7 | 99.1 |
>
>
> + **How is the model trained on graph isomorphism test datasets?**
>
>     We follow Balcilar et al. (2021) in the graph isomorphism test experiments. We label the non-isomorphic graphs with different labels and train with L2 loss. During testing, we measure the distance between pairs of graph embeddings and predict isomorphism based on a set minimum threshold distance.
>
>
> + **Larger number of IR steps does not necessarily lead to a higher classification accuracy.**
>
>     Larger number of IR steps may introduce more randomness into the embeddings, and hence, for some datasets,  this may affect generalization. In such cases, a larger value of K may be required to get embeddings with less variance. Table 7 results also suggest that for a fixed T, accuracy increases with increasing number of paths, K. Note that larger T helps in breaking the symmetry and larger K helps in reducing variance. In most real datasets, a small value of T is good enough to get the required expressivity.
>
> References:
>
> Kriege et al. (2018) A Property Testing Framework for the Theoretical Expressivity of Graph Kernels. IJCAI 2018.
>
> Balcilar et al. (2021), Breaking the limits of message passing graph neural networks. ICML 2021, arXiv:2106.04319

---

> > ### Comment · Reviewer_WUrz · 2021-11-22
> > **Re**
> >
> > I would like to thank the authors for the detailed response. Most of my concerns were addressed and I am still leaning towards accepting this paper. It would be nice if the authors could evaluate the PF-GNN model on some additional real-world datasets.

---

> > > ### Author Response · Authors · 2021-11-23
> > > **Author reply**
> > >
> > > Thank you very much for your positive response to our work! We appreciate your thoughtful comments which helped us improve the paper.
> > >
> > > We will add further results on real-world datasets as suggested.Thanks again!

---

### Author Response · Authors · 2021-11-21
**General Response**


We thank all the reviewers for their time and constructive feedback! We have updated the manuscript addressing the rebuttal phase discussion with the reviewers and have uploaded the revised version. In summary, we have improved our paper with the following:
1. Page 6: Added a discussion on the number of paths (K) needed to get good embeddings with less variance.
2. Page 6: Added further justification for weights update function in the absence of  conventional intermediate observations.
3. Table 3 of section 5.2: Added results on Random Node Initialization including partial RNI.
4. Table 7 of section 5.3: Added results on OGB-molhiv dataset.
5. Theorem 2 of Appendix A.3: Added explanation from Eq (22) to Eq (23).
6. Section A.5.1: Added experimental ablation results on individualization function
7. Section A.5.2: Added experimental ablation results on Sequential individualization
8. Section A.5.3: Added additional results on Runtime analysis on real world datasets.
9. Section A.5.4: Added additional results on Graph property testing for connectivity, planarity and bipartiteness.
10. Section A.6: Added Extended Preliminaries section with additional explanation of graph coloring procedure for isomorphism and terms associated with it along with figures.
11. In order to make space, we have moved the pseudocode of Algorithm 1 to Section A.4.

We hope our modifications and clarifications have addressed most of the concerns about our paper. We would be happy to discuss and answer any further questions you may have.

---

### Decision · Program_Chairs · 2022-01-20

**Decision:**

Accept (Poster)

**Comment:**

This paper presents a neural version of individual-refinement (IR) architecture for improving the expressiveness of GNN in terms of isomorphism tests. As IR is the dominant approach of practical graph isomorphism tests, adapting IR to GNN is a novel and important idea. As IR suffers from the exponential number of branches, the paper adapts particle filtering algorithms to sample K paths to approximate the full version of the IR algorithm. Simulation and real-world datasets are used to demonstrate the improvement over base GNN.

Strengths:
+ The paper is well written and easy to follow.

+ The originality of the paper is high since it is both technically rich. Adapting individual-refinement (IR) to Neural and GNNs is a novel and important contribution. The designed algorithm does improve over base GNNs.

+ The particle filtering algorithm is an elegant and low-complexity realization of the IR algorithm.


Weaknesses:

- PF-GNN is mainly evaluated on synthetic datasets, while only three real-world datasets are employed in total. It is not thus entirely clear how effective the proposed model is in real-world scenarios. The authors added a new real-world dataset, OGB-molhiv, during the discussion period.

- The major weakness is the scalability and practical complexity. The designed model is T times deeper and larger than the base model, and K path sampling needs K times larger memory for parallel computation. Although the reviewers still have some concerns, such as “sampling method may judge isomorphic graphs as non-isomorphic”, the reviewers appreciate the author’s hard work on partially addressing the problem during the discussion period. We encourage the author to continue to improve the paper along this direction.